# The Wetlands Paradigm Shift in Response to Changing Societal Priorities: A Reflective Review

**Edward Maltby**

Department of Wetland Science, Water and Ecosystem Management, School of Environmental Sciences, University of Liverpool, Liverpool L69 3BX, UK; e.maltby@liv.ac.uk

**Abstract:** This paper reviews some of the key influences that wetlands have had on the development of human society together with the history of wetland use, conservation and management in the context of changing human interactions from prehistoric to modern times. It documents the origins of the Ramsar Convention and the changes in the criteria for defining wetlands of international importance from an emphasis on migratory birds to those of wider functional importance contributing to community well-being. This led to a significant increase in the number of signatories from developing countries The change in scientific emphasis from ecology to ecosystems (and ecosystem services) is identified as a key element of the wetland paradigm shift, which has occurred in the last half century and renewed the recognition of the importance of the natural capital of wetlands. It represents a change in research agenda from what wetlands are to what wetlands do. Modification of the Ramsar wise use concept is documented, and evolution of wetland assessment methods is traced in relation to policy development and the need for a strong science evidence base to improve decision-making connected with wetland conservation and management. The author also addresses the significance of wetland economic valuation and biodiversity issues, transboundary water management with particular reference to the marshlands of Mesopotamia (southern Iraq), conflict, and human livelihood issues. Examples are given of the drive towards wetland restoration in different countries, and at different scales, with awareness of the extraordinarily high costs associated with major schemes such as the Florida Everglades which may prohibit replication in other parts of the world. Adoption of the Ecosystem Approach and the "Wholescapes" concept are seen as important in the future management of wetland ecosystems. The wide-ranging interactions within the structure of a new wetland paradigm are summarized diagrammatically. An examination of current societal priorities and challenges resulting from the nexus of issues arising from food production, energy, water, and environmental change and health suggests both significant threats to wetlands, but also some opportunities for these ecosystems to play a part in sustainable solutions contributing to human well-being. The paper concludes with an endorsement of a new World Charter for wetlands but emphasizes the vital importance of partnership working and the key engagement of local communities to make any new initiative for enhanced protection and management of wetlands to work on the ground. Key challenges facing wetland science are identified, but it is the realization that healthy wetland ecosystems are a significant contributor to human and societal well-being that underpins the paradigm shift in research, management and policy needs.

**Keywords:** wetland management history; Ramsar convention; wise use; wetland assessment methods; wetland valuation; wetlands paradigm shift; ecosystem approach; wholescapes; sustainable development; climate change; World Charter for wetlands



## 1. Introduction: Change Is a Normal and Persistent Feature of Earth, Human and Societal History

Earth environmental conditions have evolved and been impacted by natural processes and events throughout geological time. Humans also have altered them, more or less dramatically, by deliberate actions or unintended consequences of activities since prehistory.

What is not necessarily normal or predictable is the scale, location, and speed of change. Yet we now have unprecedented levels of knowledge from ever-increasing bodies of research which can help us to adapt to, reduce, and avoid undesirable consequences of change.

Wetlands have played a seminal role in Earth history and in some of the major environmental changes that have helped shape its present-day geological structures, waters, and atmospheric composition. Human perceptions of and relationships with wetlands have changed significantly over time and from place to place. Altered perceptions resulting from advances in scientific as well as public understanding in the last century have been reflected, at least in part, in significant policy shifts at local, national, and international levels. They have been the focus of sharp conflicts between different sectors of society with contrasting views of how they should be used. Examples of the intensity of feelings are illustrated from the UK and Australia in Figure 1.

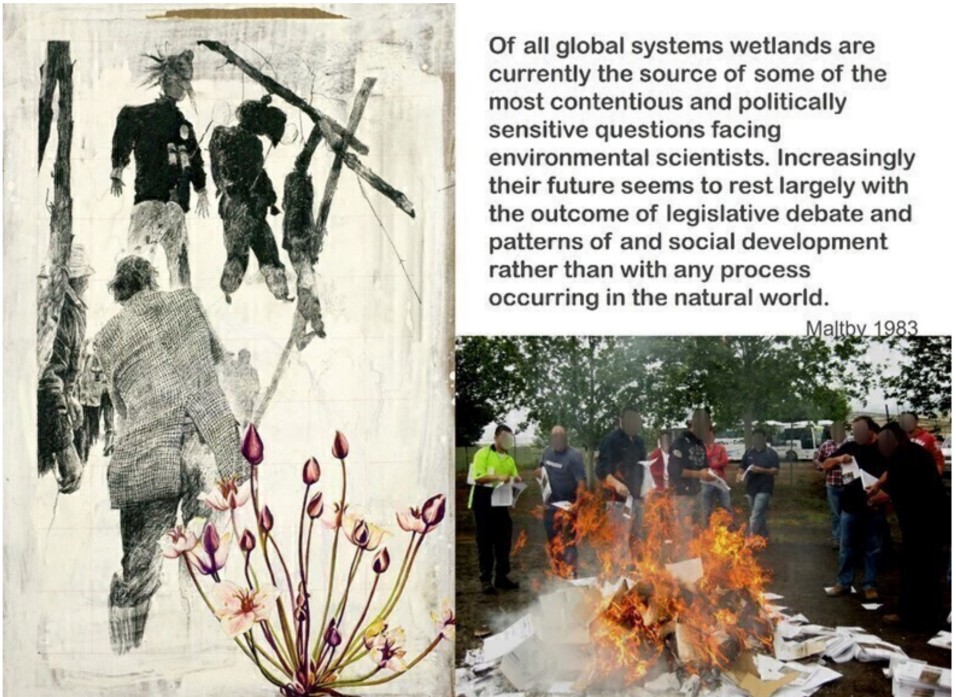

**Figure 1.** Hanging of the effigies of conservationists on the Somerset Levels, UK by farmers opposed to plans to rewet drained wetlands in 1983; Source: Jeremy Purseglove 2017 [1]. Farmers in the Murray–Darling river basin, Australia, concerned about loss of water for irrigation, burning copies of the 2010 water plan. Source: AAP/Gabrielle Dunlevy.

Over a period of some 50 years of research and advisory work, including to the Ramsar Convention, the present author has witnessed some of the extraordinary changes in our understanding of wetland ecosystems and the views that individuals and society have of them. This article offers a personal digest of some of the notable changes in perspective that have occurred, especially during this snapshot of empirical experience and attempts to interpret their significance for future generations.

Apologies are offered for the self-indulgent emphasis, but it is hoped that the experience might be insightful and in defense there are numerous excellent texts by others that are available for consultation and more comprehensive coverage.

Examples of the ways in which wetlands have been a cultural driving force are given in Maltby [2]. Setting aside their probable intimate role in the origins of life itself, wetlands have stimulated turning points in human culture by virtue of their prominence in key temporal stages or locations of Earth history. Of particular significance, given the current attention on global warming, are the tropical peat-swamp forests (such as those currently occurring in Southeast Asia (Figure 2) which were to become the extensive coal deposits of

the Carboniferous period some 250 million years ago [3]. The vulnerability of tropical peat deposits to degradation and loss due to drainage and fire is dramatically illustrated in one of the few remaining peat domes of the Mekong Delta (Figure 3).

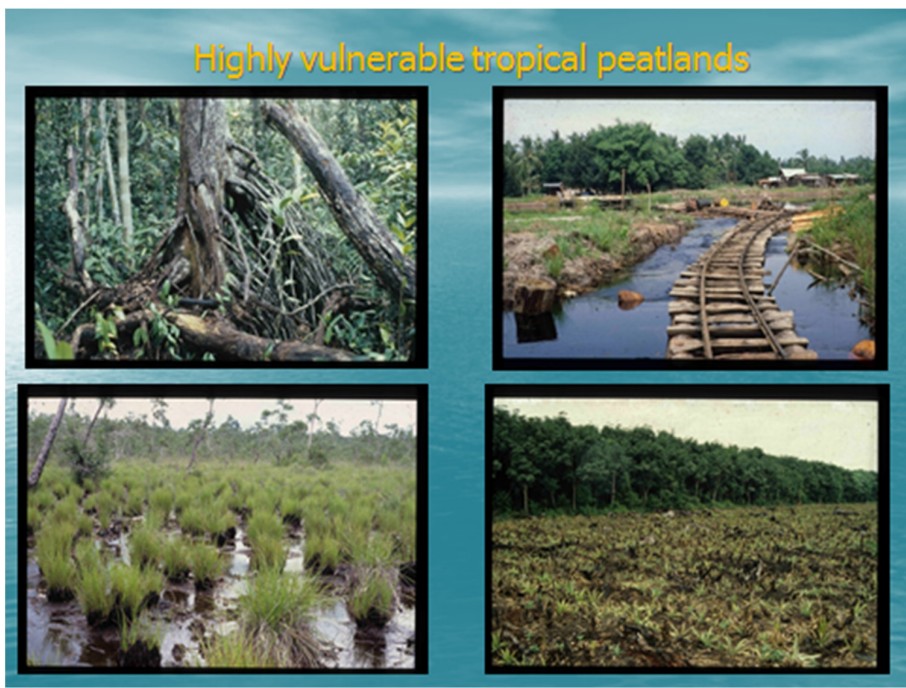

**Figure 2.** The complexity and rich biodiversity of intact peat swamp forest in Indonesia and examples of the simplification resulting from clearance and conversion to agriculture. Source: Edward Maltby.

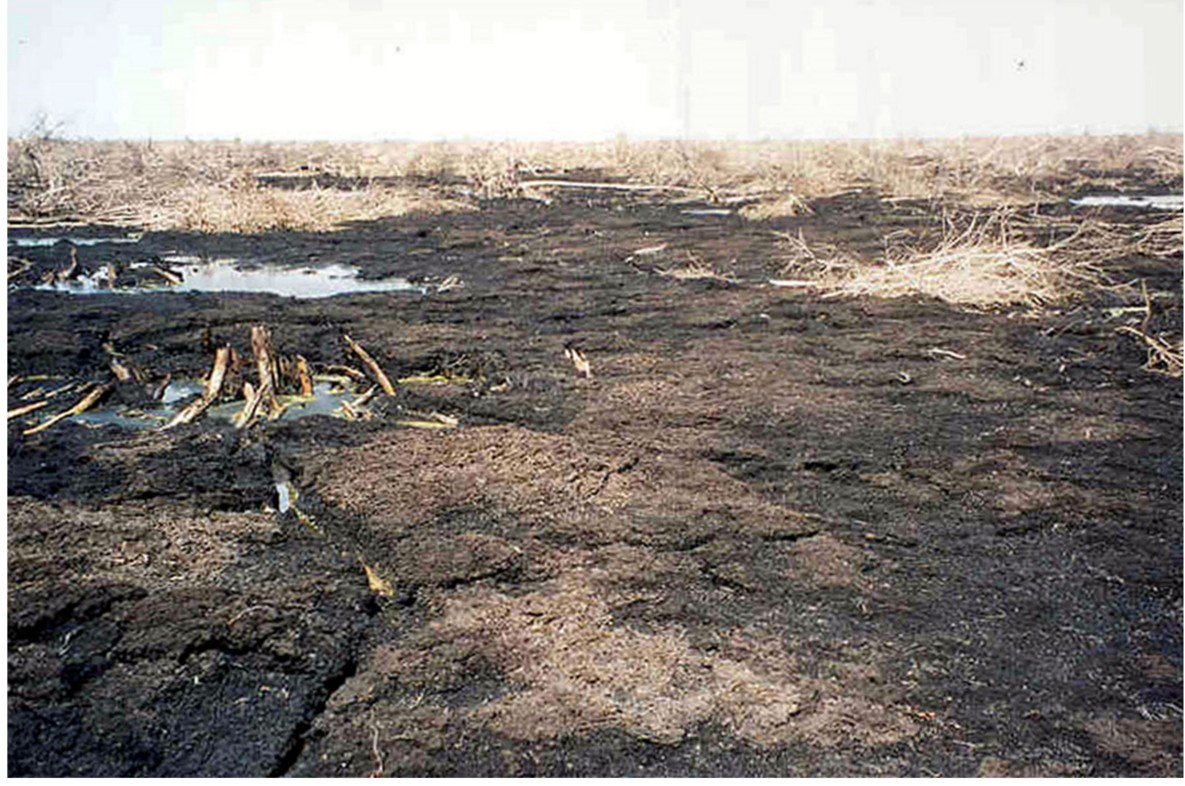

**Figure 3.** Vulnerability of tropical peatlands illustrated by the effects of fire on one of the last remaining peat masses in the Mekong Delta at U Minh Tong, Vietnam. Source: Edward Maltby.

Continental drift repositioned these vast carbon stores into higher latitudes where from the mid-eighteenth century they powered the Industrial Revolution, first in Europe and then North America. What followed was arguably one of the most significant changes in societal organization, trade and economics which would have a progressive influence not only on human society but also on our environment. Industrialization and increased national and international trading accelerated urbanization of the human population and increased the development of transport networks and hubs such as ports and centres of manufacturing. Inevitably this led to the conversion of large areas of wetland to alternative uses that has continued to more recent times (Figure 4) In effect, widespread loss and degradation of wetlands throughout the so-called developed world was the price paid (invariably never fully recognized) for rapidly accumulated economic wealth. The fact that the wealth was not shared well with a growing population and its generation also resulted in severe pollution and human health issues is now more seriously viewed in the context of our remaining 'natural capital'.

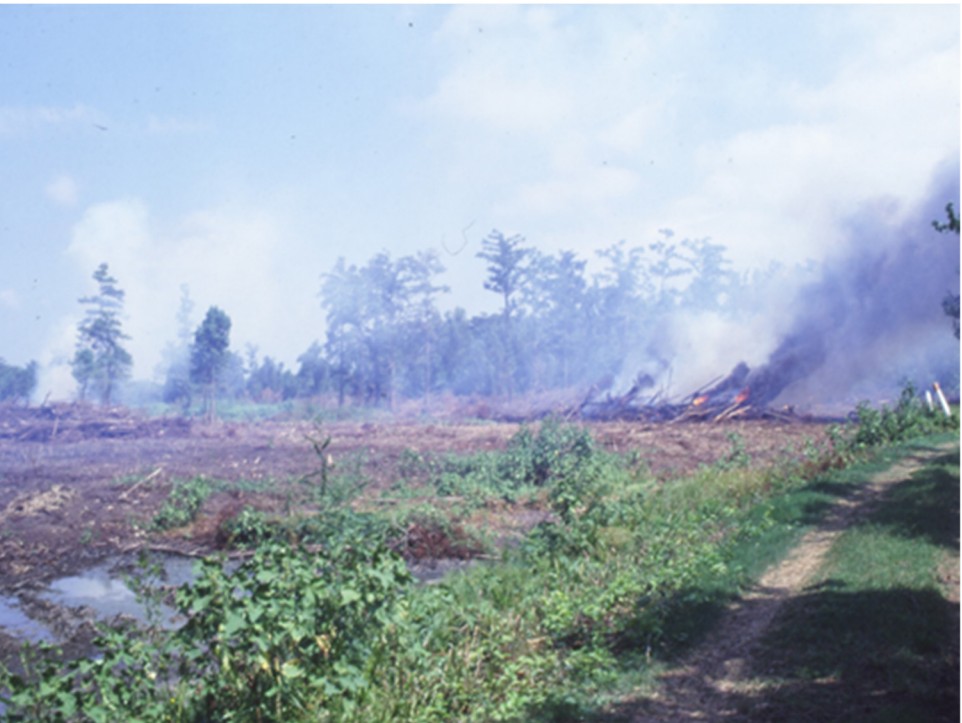

**Figure 4.** Clearance and burning of bottomland hardwood forests to convert to agricultural land in Louisiana, United States. Source: Edward Maltby.

Ironically the use of the fossil carbon stores of the wetland ecosystems from earlier geological times became a powerful direct driver of wetland loss. The wider significance of the losses was rarely recognized by the entrepreneurs and participating communities preoccupied with financial wealth creation and direct socio-economic well-being.

There was limited recognition of the adverse consequences of water and atmospheric pollution and no appreciation of the impact that the progressive release of carbon dioxide would have on climate change. It is a further irony that the reconversion to carbon dioxide of carbon previously captured by ancient wetlands should be a major contributor to accelerated global warming. The importance of the biogeochemical coupling between wetlands and larger Earth systems is now increasingly recognized as a key consideration in environmental management [2].

## 2. From Pre-History to Historical Attitudes

### 2.1. Intrinsic Dependence to Progressive Detachment

Significant evolution of the human brain is attributed by the "aquatic ape" theory to the fatty acid-rich diet available to hominins living in, and dependent on, the wetland margins of lakes, rivers and the sea [4]. Wetlands continued to play important roles in the evolution of prehistoric communities and were the sites of some of the earliest stages in the development of tool-producing hominins. Excavations of Mesolithic and Neolithic post-glacial lake marginal settlements across Europe have revealed the highly dependent relationships between humans and the wetland ecosystems [5]. Figure 5 illustrates reconstructions of community life and dependence on the wetland resources from archaeological excavations of one such settlement at Starr Carr in North Yorkshire, UK.

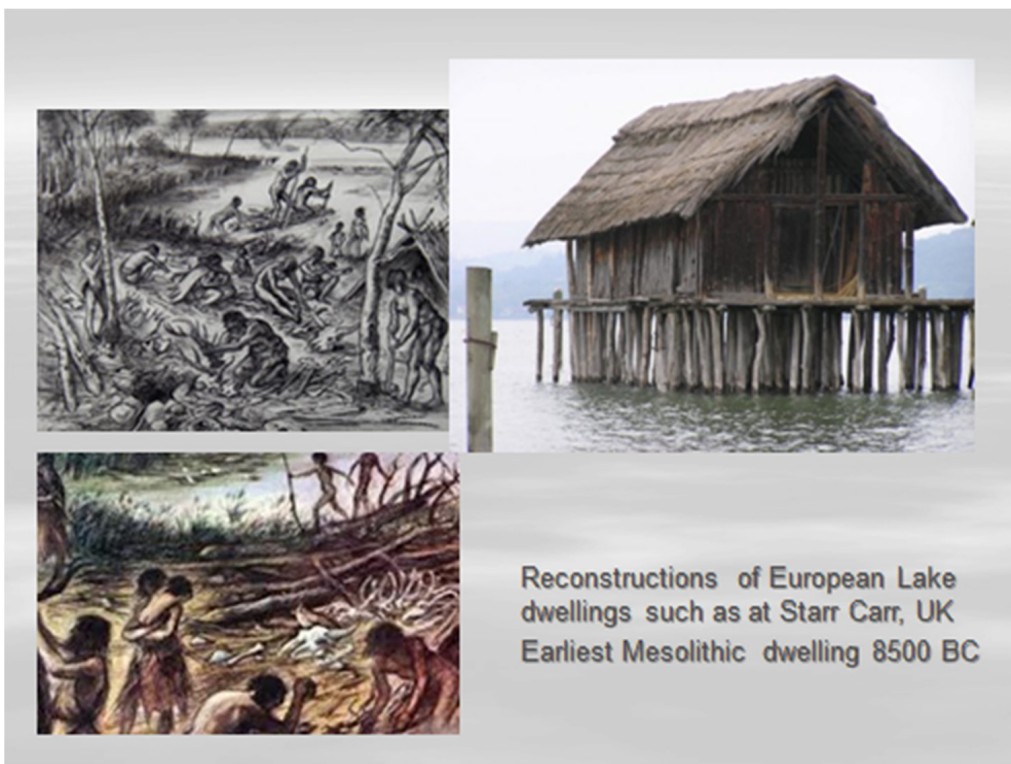

**Figure 5.** Early prehistoric communities were completely reliant on lake and river marginal wetlands throughout Europe for food, shelter, and security. Source: https://www.google.com/search?q=Star+Carr+reconstruction&sa=X&rlz=1C1ONGR_enGB1009GB1009&biw=1280&bih=569&sxsrf=ALiCzsaiGhDJ8Nv2D2DP (accessed on 26 March 2013).

Even whilst this prehistoric dependency persisted in the post-glacial communities of Europe, cradles of civilization were developing based largely on the innovative management of the waters of fertile floodplains such as those of the Nile, Tigris-Euphrates (Mesopotamia), Indo-Gangetic and North China Plains. Utilisation of the natural flood cycle created food security and the creation of agricultural wealth. It demanded community organization and cooperation together with rules and laws [6]. Cities arose from an increasingly prosperous "hydraulic civilization" [7] with food surpluses and trade supporting population growth that made possible the differentiation of roles in society that are familiar today [2]. Archaeological evidence from the city of Ur in Mesopotamia (present day Iraq) has revealed continuity in some of the basic units of floodplain life such as the reed-house (mudhif) or long canoe (mashuf) from Sumerian times to the present day—a cultural connection of more than 5000 years (Figure 6).

Some of the earliest examples of written language come from Sumer, including the origins of the biblical account of the Creation and the Flood of Noah which may be from

the epic of Gilgamesh in Sumerian literature. Such a bequest to the culture and philosophy of subsequent generations is hard to over-state.

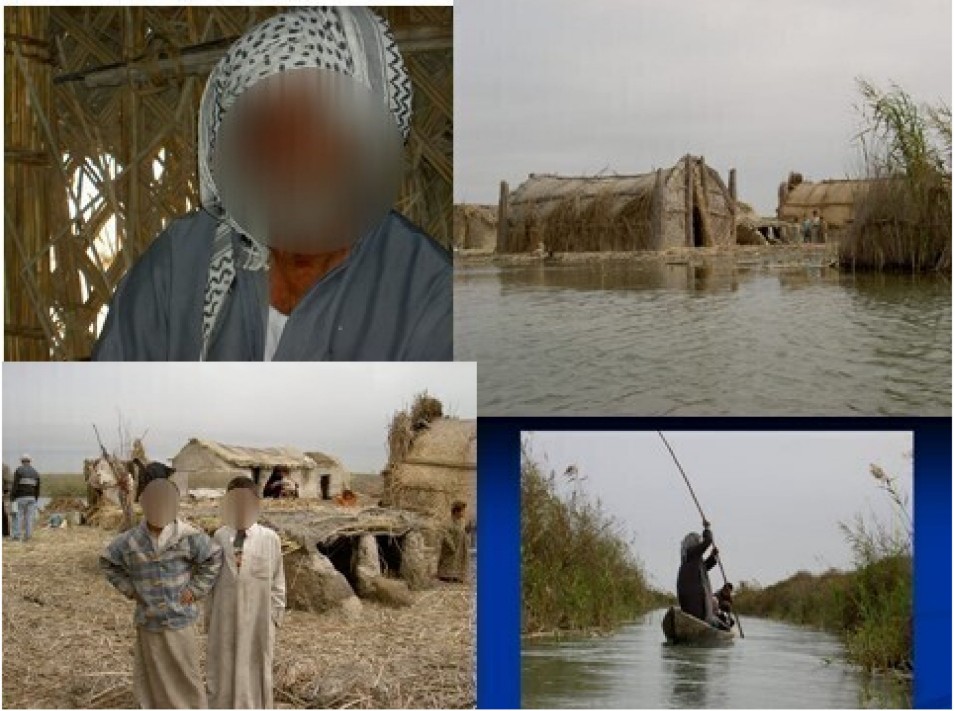

**Figure 6.** Present-day Marsh Arabs (Madan) in Southern Iraq (former Mesopotamia) with traditional reed houses (mudhif) often on floating islands and long canoe (mashuf) in a channel through marshland reflooded after 2003. Source: Edward Maltby.

The wider implications of altered hydrology of the floodplain were well understood in ancient Mesopotamia, and recognition of the enrichment of drainage waters with salt from irrigated lands was recorded as early as 4400 BC on clay tablets in the temple of Lagash. These recorded a thirty-fold increase in salinity of one particular agricultural field in just one year [8]. So was heralded the serious water quality issues which may arise from conversion of natural flood-cycle wetlands to agriculture.

Nevertheless, historical attitudes to wetlands progressively detached them from a central role in community life support. The depiction of swamps and bogs in fictional literature as sources of mythical and alien species, as well as numerous hazards to human health and safety, was highlighted by Mitsch and Gosselink [9]. Such portrayal undoubtedly influenced generations of schoolchildren throughout the developed world and became embedded in future attitudes [10]. Their drainage and transformation to other non-wetland uses were seen as laudable "public-spirited" objectives and testimony to the power of technological developments and human ingenuity [11]. We will see this view substantially reversed at the start of the twenty-first century (Figure 7).

Throughout history the benefits that we presume were appreciated by early human cultures and more certainly current traditional users of wetlands were either ignored or dismissed as less significant by more powerful sectoral interest groups [2]. Celebration of major drainage achievements such as those which empoldered the complex Rhine–Meuse–Scheldt delta from medieval to modern times; the drainage of the English Fens [12]; the more recent drainage of the Hula swamps in Israel (Figure 8); and the desiccation of large tracts of the Florida Everglades [13,14] were based on the winning of rich agricultural land, flood protection of encroaching settlement and eradication of disease vectors. The pressures for conversion have been especially strong in developing countries where the attraction of foreign earnings have often overwhelmed conservation interests (Figure 9).

European colonization of the United States resulted in widespread efforts to drain the swamps which were originally owned by the Federal government; petitions to Congress were brought seeking compensation for "improvements" undertaken by States. The resulting legislation—the Federal Swamp Land Acts of 1849—50 and 1860—was intended to reduce flood hazard, improve sanitation and reclaim land for agriculture [15]. For centuries "the drainage of wetlands has been seen as a progressive public-spirited endeavour. the very antithesis of vandalism" [11]. Farmers and other landowners were encouraged by the availability of generous government grants and tax concessions in the UK and elsewhere to convert wetlands to more productive agriculture and other land uses such as forestry.

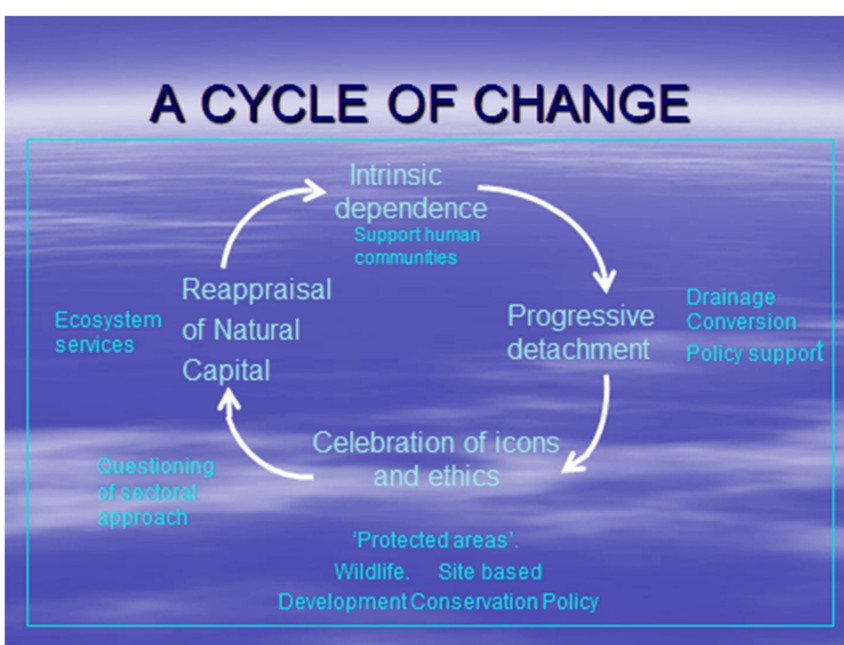

**Figure 7.** A simplified and generalized representation of the circle of change of the human interaction with wetlands. Source: Edward Maltby.

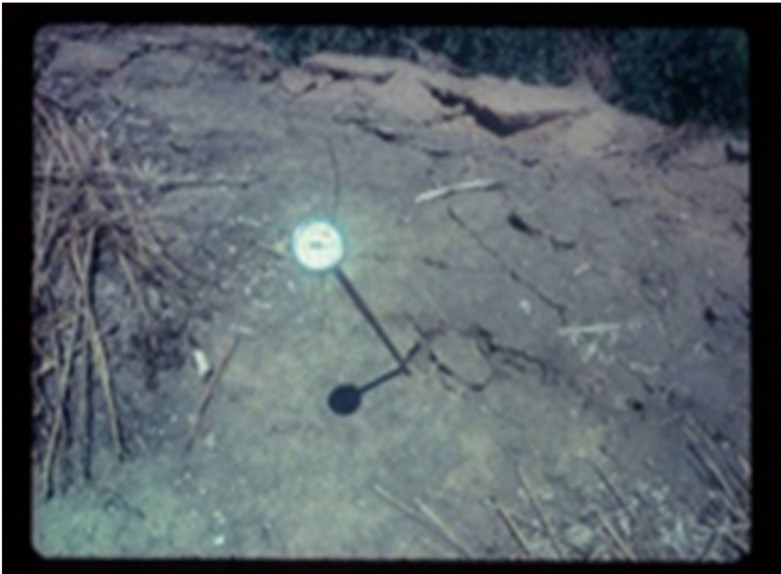

**Figure 8.** Peat fires at depth were a common result of the drainage of the Hula valley swamps in Israel. The temperature probe in the residual ash reveals prolonged sub-surface burning. Source: Edward Maltby.

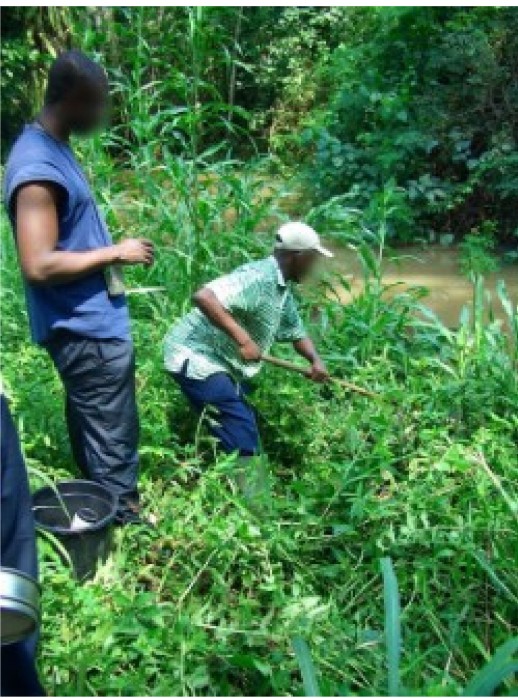

**Figure 9.** The rich biodiversity of stream, river and lake marginal wetlands throughout Africa, such as this example from Ghana, are under constant threat from unsustainable water extraction and landuse pressures to meet increasing economic needs and poverty alleviation. Source: Edward Maltby.

*2.2. Celebration of Ecology and the Rise of Conservation Icons and Policy*

Early ecological research focused on habitats and species which only much later became grouped under the umbrella term "wetlands". Studies were of bogs, fens, carr woodland, mires, muskeg swamp, marshes or other habitat descriptors without any reference to "wetland". Figure 10 illustrates just some of the range of such habitats and species.

The term *wetland* was first used in 1953, in a report by the U.S. Fish and Wildlife Service (USFWS) that provided a framework for the first official use of the term in a later publication concerning waterfowl habitat in the United States [16]. It may be that the specific link to waterfowl habitat, as well as the limited interaction between scientists in the United States and those from other countries at this time, explains the reluctance to immediately accept the term beyond North America. Indeed, numerous well-respected European scientists refused for many years to use the term in their empirical research and publications, primarily because they argued that it was too generic and grouped together such highly diverse ecological systems as to become taxonomically confusing. Nevertheless, it became embodied in the only multinational treaty to identify with a single group of ecosystems—the Ramsar Convention.

Early emphasis in research was on fundamental ecological questions. Such can be exemplified by the investigation of the origins, diversity and associated flora and fauna of peat-forming systems, e.g., Moore [17], Moore and Bellamy [18]. They are a pertinent choice given the current switch of focus from basic ecology to their delivery of important ecosystem services and especially their role in greenhouse gas dynamics and climate change. The "formation, development and maintenance of peat depends on an envelope of environmental conditions representing local- and/or regional-scale characteristics in which climate is of critical but not sole importance" (cf. templates of formation) [18]. It has long been stated that peat-forming ecosystems in the UK uplands were originally primarily the result of climate-topographic interactions [18]. A correlation has been made between the 1200 to 1250 mm isohyet (line of equal precipitation) and the extent of blanket peat [19,20], whereas Rodwell [21] reported the threshold for blanket peat formation as at least 160 rain days combined with annual rainfall greater than 1200 mm. Lindsay et al. [22] correlated

the change in size, proportion and pattern of open pools and hollows in blanket bog ecosystems with the number of rain days and mean temperature. Much less attention was given originally by ecologists to the potentially overwhelming importance of prehistoric human activities to the onset and subsequent development of upland peat. It is now accepted that anthropogenic activity has played an important role in the development of blanket peat over a substantial area of the UK [17,23,24] through clearance of woodland and hydrological alterations at a time when climate overall was favourable for the growth of *Sphagnum* mosses [25].

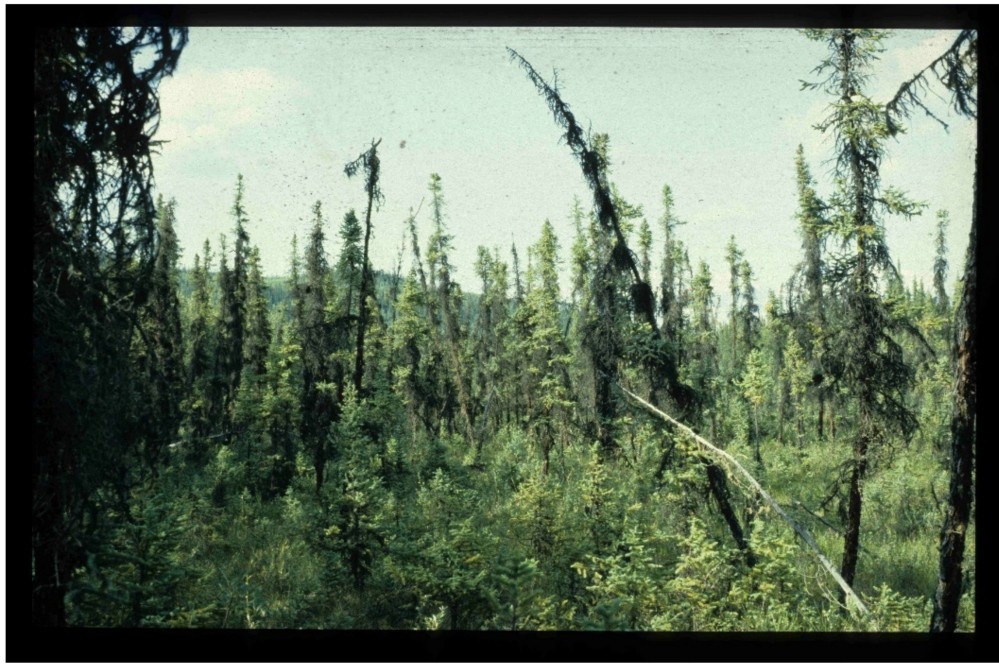

(**a**)

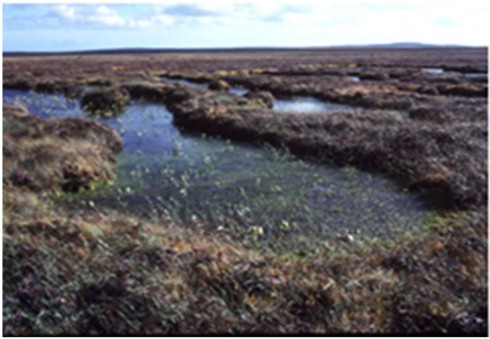

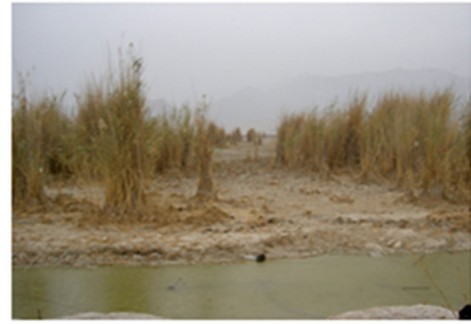

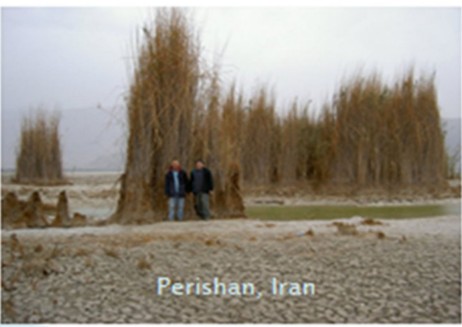

(**b**)

**Figure 10.** *Cont.*

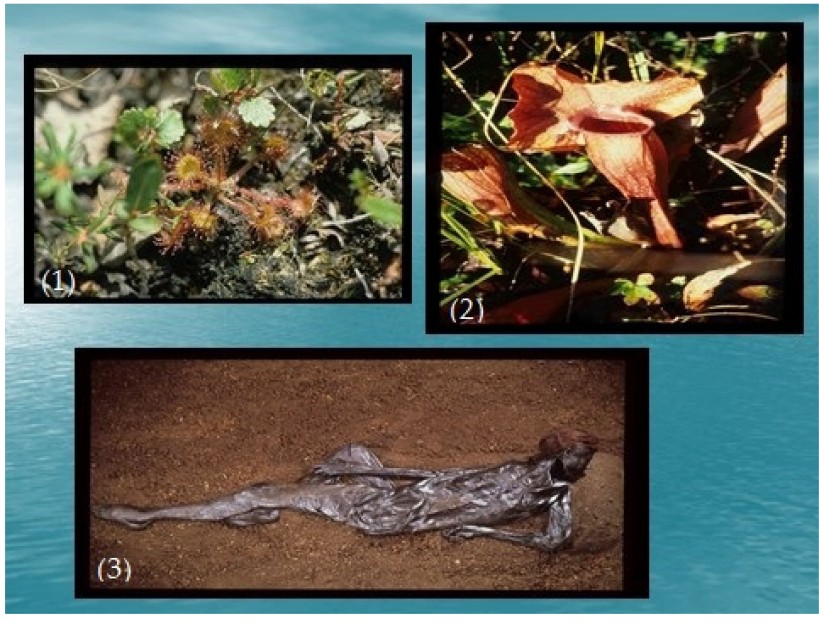

(c)

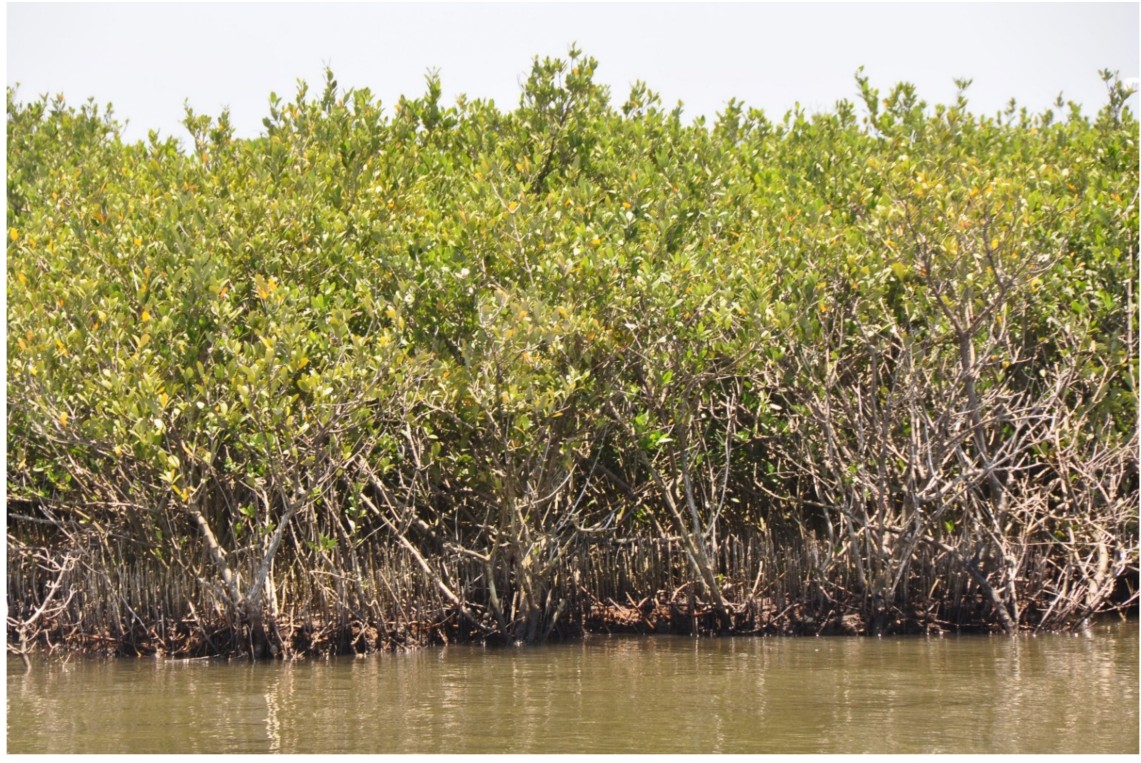

(d)

**Figure 10.** *Cont.*

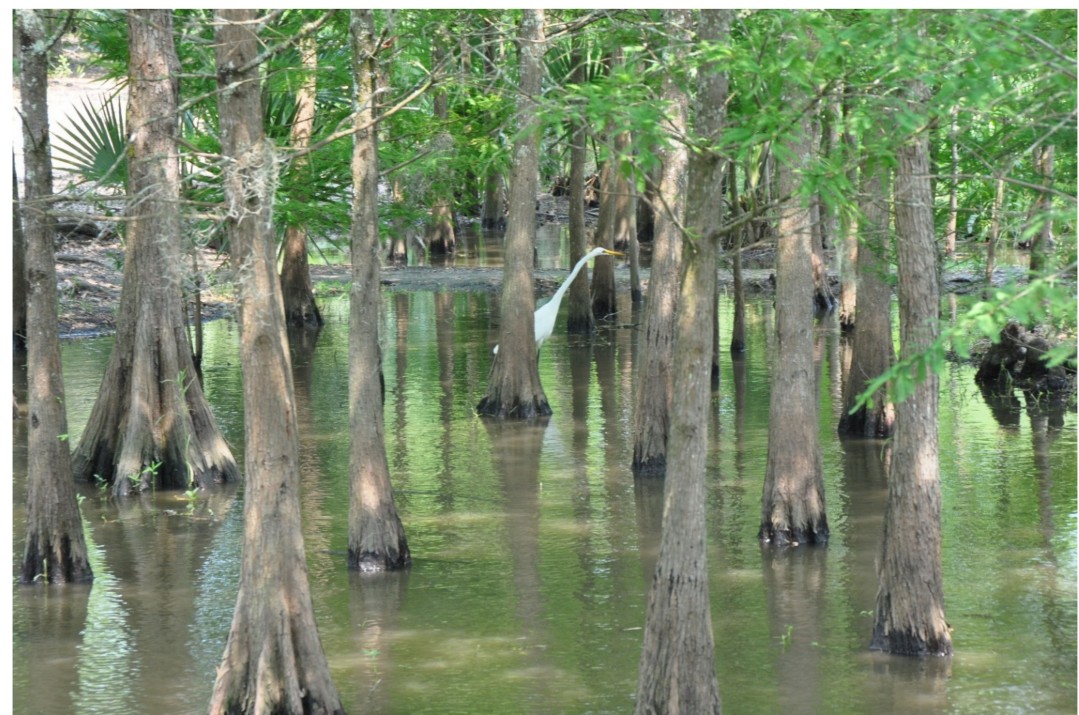

(**e**)

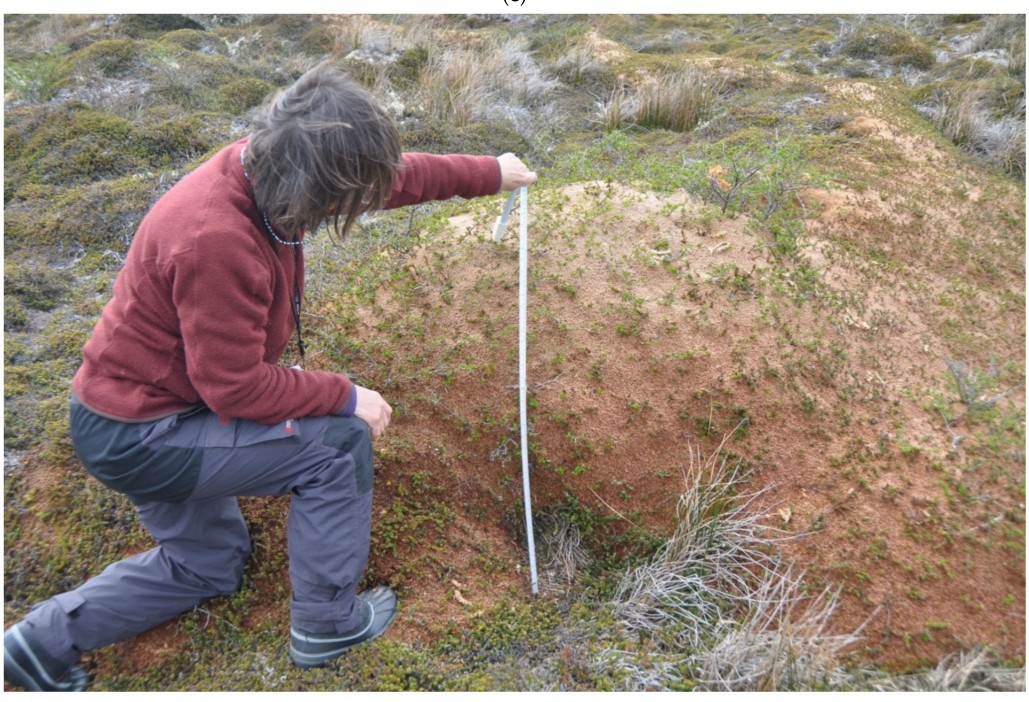

(**f**)

**Figure 10.** Selection of the extreme diversity of wetland types and species: (**a**) muskeg in Alaska showing "drunken" stand of Black spruce (*Picea mariana*) on peat and permafrost; (**b**) the distinctive microrelief patterning of hummock and pools in "hollows" of blanket bog and the extraordinary height of stands of Phragmites in the Perishan wetland of Iran; (**c**) examples of carnivorous plants—(**1**) sundew and (**2**) pitcher plants—adapted to source nutrients from insects to supplement nutrient deficiencies in the peaty substrate, in both temperate and tropical climates (**3**) Human bodies uniquely preserved in the acidic waterlogged peat provide evidence of diet, associated environmental conditions as well as cause of death; (**d**) fringing mangroves in coastal Louisiana; (**e**) cypress swamp (*Taxodium distichum*) in Florida; (**f**) Hummocks of *Spagnum magellanicum* in Tierra del Fuego bog. Source: Edward Maltby.

Somewhat ironically, the actual trigger for peat development might have been human-induced alteration of vegetation and/or soil conditions such as the development of an impermeable iron pan in podzols resulting in hydrological change [26] without necessarily requiring climate change. Dimbleby [27] had drawn attention in the UK to human-induced podzolisation in the Bronze Age stimulating the development of peaty soils, whereas Maltby and Caseldine [28] provided direct pedogenic, pollen and 14C evidence for the possible high speed of change from brown soils to those accumulating a strongly acid peat surface in the Bronze Age (ca. 3500 BP) on Bodmin Moor (Figure 11).

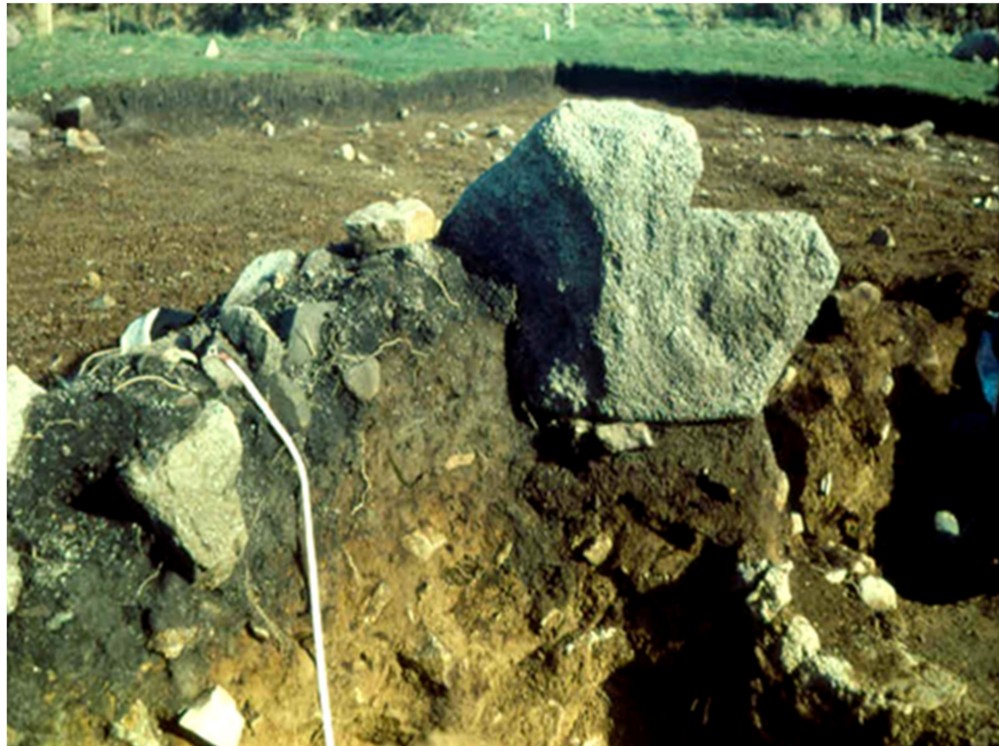

**Figure 11.** Original brown earth type forest soil preserved beneath a Bronze Age burial structure with a thin iron-pan podzol and peaty surface developed in the surface of the structure and peat accumulated in the surrounding area; Colliford, Bodmin Moor, southeast England. Source: Edward Maltby.

Research highlighted the particular significance of hydrology and water movement in determining the stability, size and slope of the peat mass [29,30]. The natural limits to peat growth, the overall carbon budget and the physical stability of the peat mass was elegantly demonstrated by Clymo [31,32].

Such early ecological research across diverse habitat types played a vital role in our understanding of what wetlands were, how they developed, and why they occurred where they did; this does not mean that we have all the answers to some of the basic ecological questions. The accumulation of deep peat under the remarkably dry climatic conditions of the Falkland Islands, for example, is difficult to explain on the basis of the understanding of peat bog formation in the Northern Hemisphere [33]. Figures 12 and 13 illustrate the remarkable depth and formations of peat on East Falkland, notwithstanding the relatively dry climate.

The sequential accumulation of organic matter and sediment provided palaeoecologists with a time machine from which to document environmental change as well as evolution of the wetland itself.

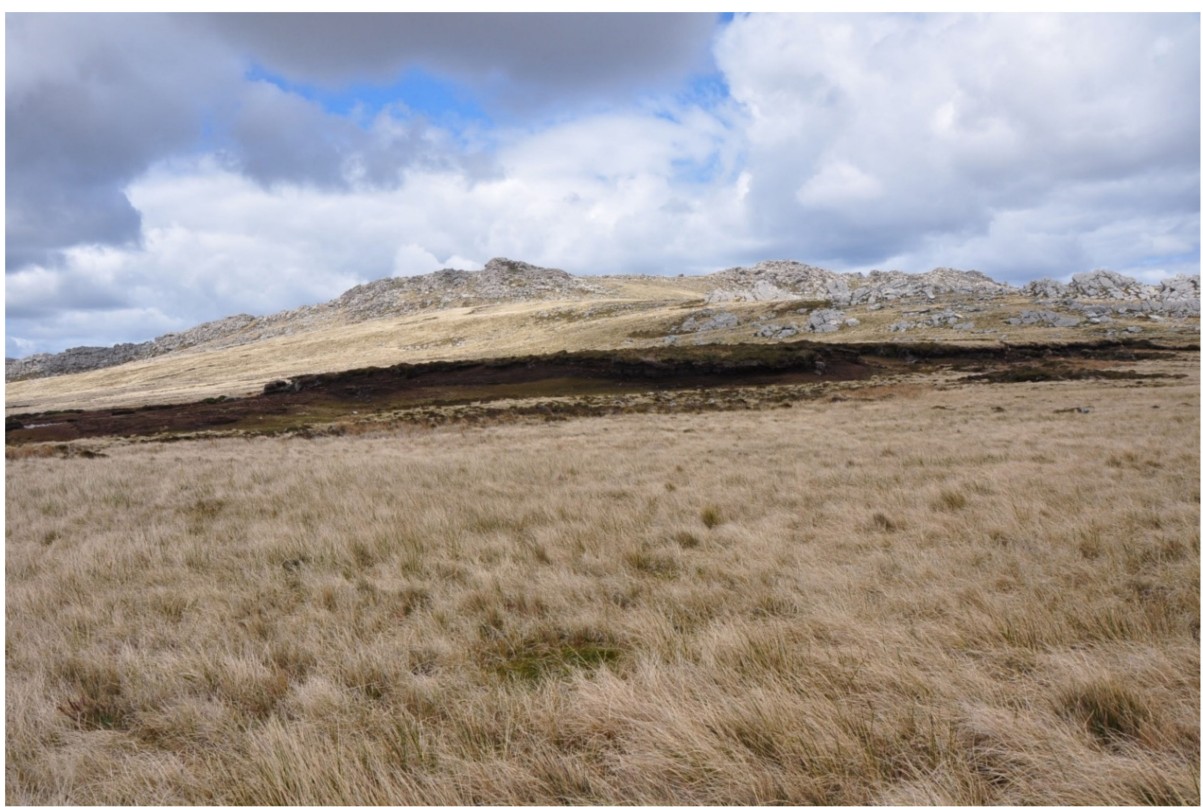

**Figure 12.** View of Falkland Island peat bank (bog) between Mt Harriet and Goat Ridge. Source: Edward Maltby.

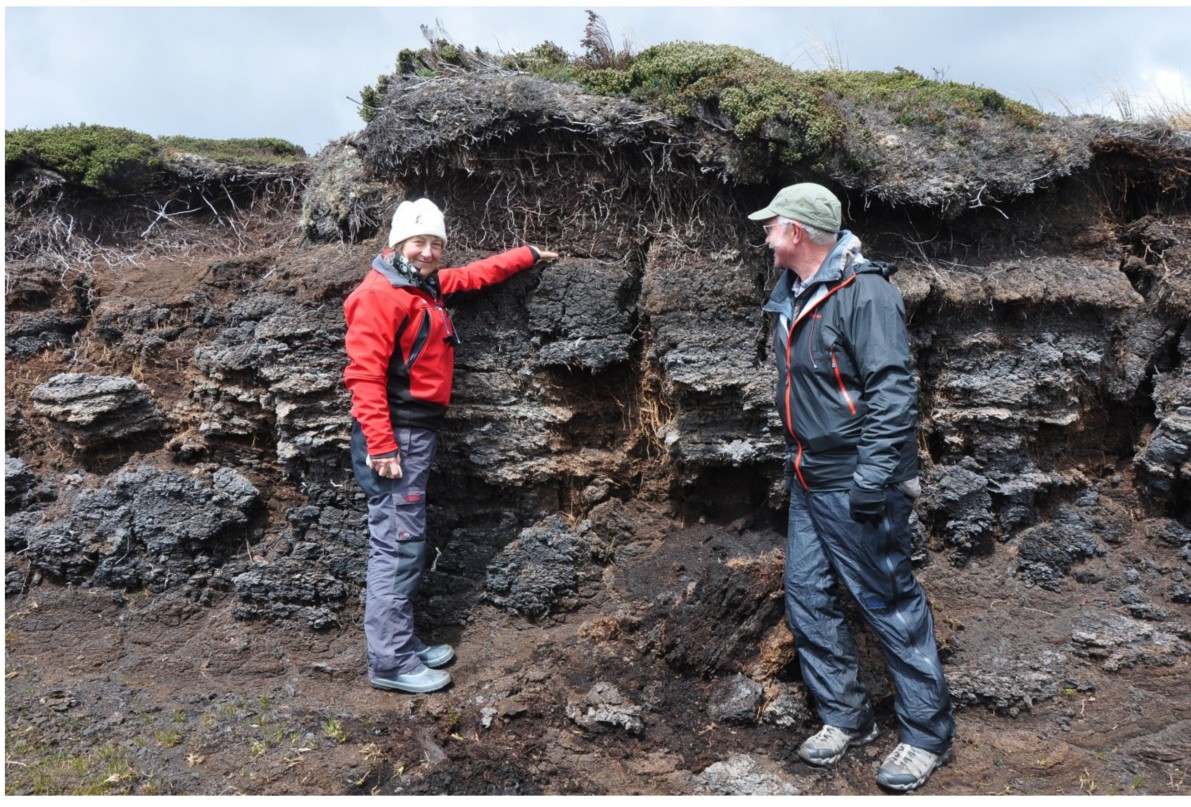

**Figure 13.** Exposed eroding face of the peat 'bank' at Goat Ridge, Falkland Islands. Source: Edward Maltby.

Maltby [2] identified characteristics of habitats now described as wetlands which spawned examples of important areas of ecological understanding, notably:

1.  Change over time through ecological succession—frequently they are transient features in the landscape and in many cases may be regarded as authors of their own destruction;
2.  Zonation both within and at the boundary of the wetland is a common feature of species distribution and substrate characteristics;
3.  Temporal variations through seasonal or episodic hydrological or flood "pulsing" cycles—see especially Junk et al. [34] or individual events.

Initially research was essentially site-based and has contributed to a wealth of taxonomic data and understanding of species-habitat relationships.

Inevitably the focus on wetlands as parts of the landscape worthy and in need of protection also became site based as well as their importance for particular species and especially birds. Indeed, this was also reflected in the central commitment to the Ramsar Convention which required signatories to list at least one wetland site of International Importance.

The growing concern for wildlife and the vulnerability of their habitats had already begun to emerge at the end of the nineteenth century. In the UK the Royal Society for the Protection of Birds (RSPB) was formed in 1889 and was highly influential in the passing of the Bird Protection Act in 1954 (much earlier a Wild Birds Protection Act had been passed in 1872).

Particularly significant in the United States was the founding in 1886 of the National Audubon Society in response to the unprecedented levels of hunting, particularly of wetland birds for their plumage that was much sought after to satisfy the demands of the millinery trade responding to the fashion requirements of an increasingly wealthy part of Society. For some 50 years from about 1870, plume hunters invaded the Everglades and in particular decimated colonies of the Snowy Egret that was much favoured by the fashion houses of the day. In the mid-1880s up to five million birds a year were being harvested just for their plumage [35]. The Florida Audubon Society requested the American Ornithologists Union to hire a game warden to prevent the illegal killing. Guy Bradley was the first such warden in South Florida who was tragically killed in 1905 trying to arrest a notorious plume hunter. His death further inspired the conservation movement and was an additional stimulus for future legislation. Although birds were a key driver of the conservation movement, there are many plants and other animals that are unique and charismatic features of wetlands. The American alligator is used as an example of the specialized fauna, not least because of its ancient origins in the Oligocene period some 30 million years ago (Figure 14).

The year 1905 was also significant in the Netherlands where the Society for Preservation of Nature Monuments was established in response to plans by Amsterdam City Council to use a nearby wetland as a rubbish dump for the rapidly expanding city; the Society bought the wetland and saved it from loss. Today Natuur Monumenten is one of the largest landowners of nature reserves and parks in the Netherlands (where of course much of the country historically was wetland).

Whilst wetland species and specific iconic sites were often the *cause celebre* for the early conservationists, a more general environmental movement achieved prominence only from the 1960s with many crediting Rachel Carson's book *Silent Spring*, published in 1962 [36], as an influential catalyst. Her message was important in demonstrating the devastating impacts of pesticides on wild non-target species often via the links provided by water. Recognition of the coupling among environmental contaminants, food chains and the water cycle was a precursor of the subsequent paradigm shift that would take us to the present day.

In the middle of the twentieth century the conservation focus was not only site and species based, but primarily concerned with the preservation of rarity, uniqueness or particularly good examples of nature. Following its much earlier connection with the inspiration for conservation, Everglades National Park was officially opened in 1947 when land and funding was eventually secured and became the largest designated subtropical

wilderness reserve on the North American continent. It became an International Biosphere Reserve in 1976, a World Heritage Site in 1979 and a Ramsar site in 1987, to confirm its position as one of the world's iconic wetlands.

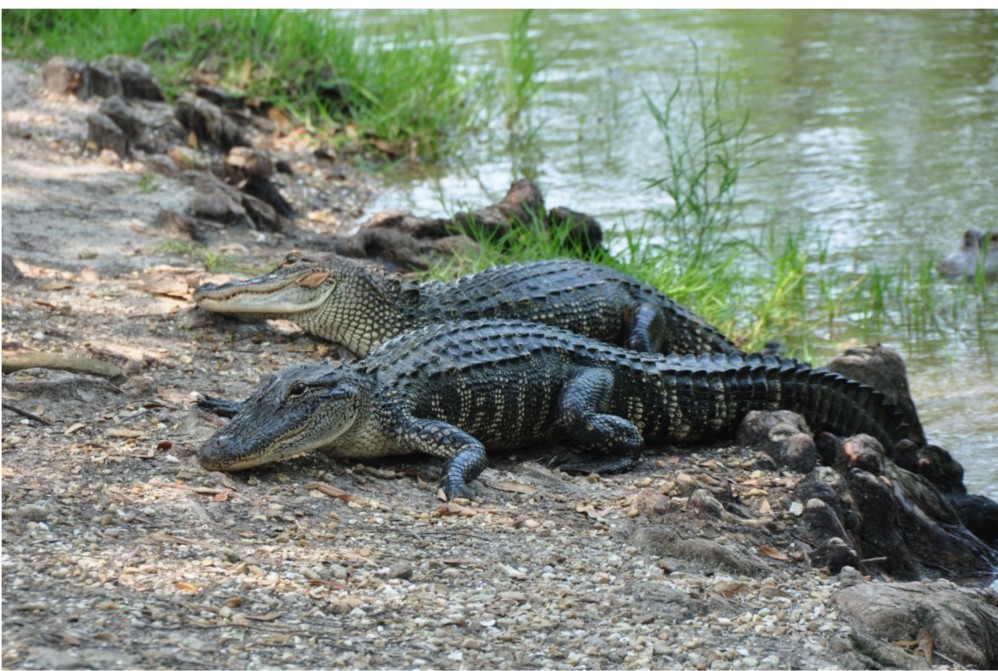

**Figure 14.** Example of American alligators in their natural wetland habitat of Louisiana. Source: Edward Maltby.

Mid twentieth-century threats to wetlands resulting from sectoral policy conflicts also managed to galvanize communities around the conservation banner. In the UK the Forestry Commission was established by government to enhance the much-depleted post-First World War forest stock. It later proposed (in the 1950s) to plant coniferous trees on The Chains, the most extensive tract of blanket bog on Exmoor. At the same time there was substantial government grant aid to support the conversion of peat and organic soils to more productive farmland. In the case of Exmoor this led to the formation (in 1958) of the Exmoor Society, an effective non-governmental organisation (NGO) and lobby group dedicated to maintaining the "traditional" moorland landscape (Figure 15) and its associated rural livelihoods and public access. Afforestation on Exmoor was prevented, the blanket bogs became a celebrated conservation icon, and the Exmoor Society a beacon for policy change.

Policy innovations included management agreements with landowners who were compensated to prevent alteration of the moorland landscape. It is ironical that much of the moorland so cherished at the present time was covered in mixed deciduous woodland in prehistoric times; clearance, especially during the Bronze Age by human communities, served as a catalyst for ecological and pedogenic change resulting in peat development. For immediately succeeding generations this impact might have been considered as a first environmental planning disaster of that period because of the decline in sources of food, shelter and heating [25].

The blanket peats of Caithness and Sutherland in Scotland were not as fortunate as The Chains, and 67,000 ha (17%) of the peatland were either planted or approved for planting [37] as a result of tax incentives or Forest Grant Scheme approval [22]. The tax incentives have since disappeared and the Forestry Commission now has guidelines to protect blanket bog from further afforestation [38].

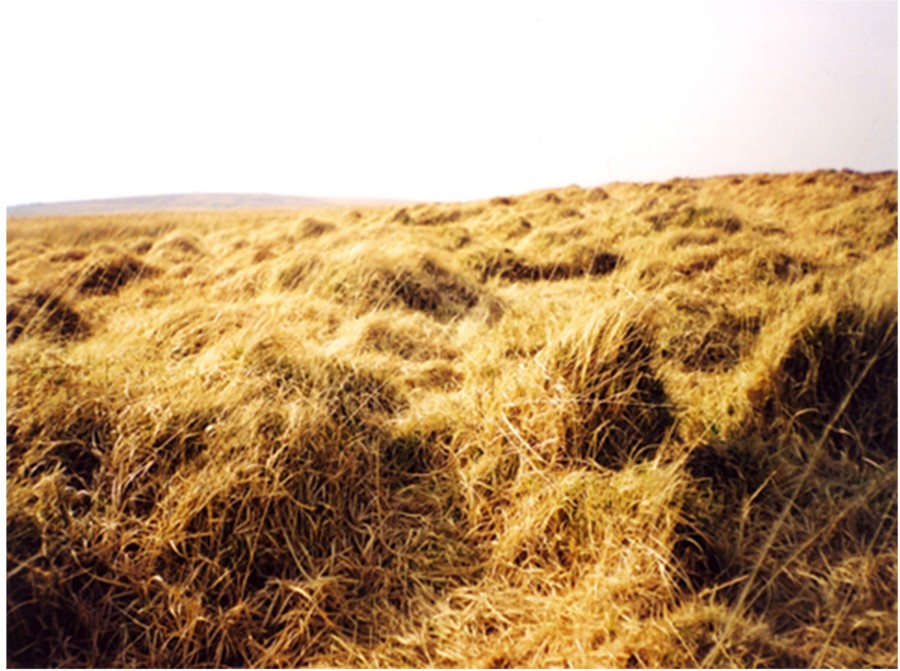

**Figure 15.** The Chains blanket bog, Exmoor, UK; dominated over large tracts by the deciduous grass *Molinia caerulea* which has contributed to active peat accumulation and was threatened by afforestation. Source: Edward Maltby.

Individual site-based conservation is enshrined in the various scales of protected area policies worldwide. Within the UK the Wildlife and Countryside Act of 1981 empowered the government conservation body, Natural England, to identify and protect Sites of Special Scientific Interest (SSSIs) on the basis of wildlife, geology or landform. These include numerous wetlands, but protection by law of the site does not necessarily ensure protection from external impacts.

The Ramsar Convention in 1971 provided a seminal change in the conservation strategy for wetlands by recognizing the importance for migratory birds of not just single sites, but of the network connections of wintering, breeding and feeding (especially re-fueling and resting sites) often over hundreds if not thousands of kilometres and across multiple sovereign territories. In recognition of the rapid and accelerating losses of wetland habitats, especially throughout Europe and North America, the movement to secure conservation and where possible restoration of the remaining resource at the global scale gained traction. The Ramsar Convention became an iconic standard bearer for the conservation and sound management of wetlands. It emerged from the efforts of passionate ornithologists, notably Luc Hoffman from Tour du Valat research station in the Camargue, France, Geoffrey Matthews from The Wildfowl and Wetlands Trust in Slimbridge, UK, and Escandar Firouz, former Minister of Environment in Iran. The key motivation was recognition that it was necessary to safeguard the essential international connectivity of habitat requirements of migratory waterfowl. Single site conservation and management alone was not sufficient to maintain the populations of migratory species which weremuch revered by bird-watchers throughout the developed world. The Ramsar Convention on Wetlands of International Importance Especially as Waterfowl Habitat was established in 1971 at a meeting in Ramsar on the shores of the Caspian Sea (Iran) and became unique as being the only international agreement to cover a specific single group of ecosystem type.

## 3. From Ecology to Ecosystems and Ecosystem Services

The last 50 years has witnessed a major shift in scientific research from the description of wetland habitats and their relationships with plants and animals to attempting to

understand how wetland ecosystems actually work, as part of larger environmental systems and in support of human well-being and exemplified in Figures 16 and 17.

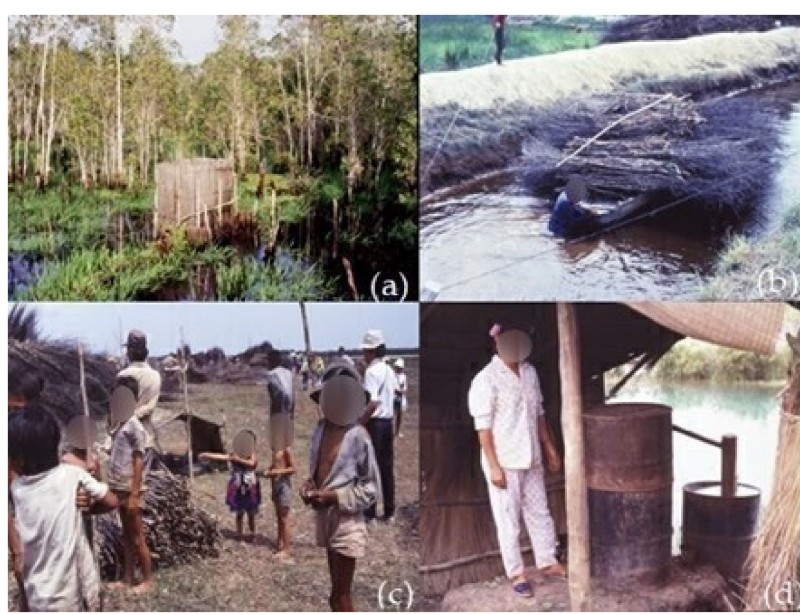

**Figure 16.** Examples of the services realized by communities in the natural Melaleuca wetlands of the Mekong Delta, Vietnam; (**a**) fish trap; (**b**) cut Melaleuca stems; (**c**) sorting wood for different uses; (**d**) distillation of essential oils from leaves. Source: Edward Maltby.

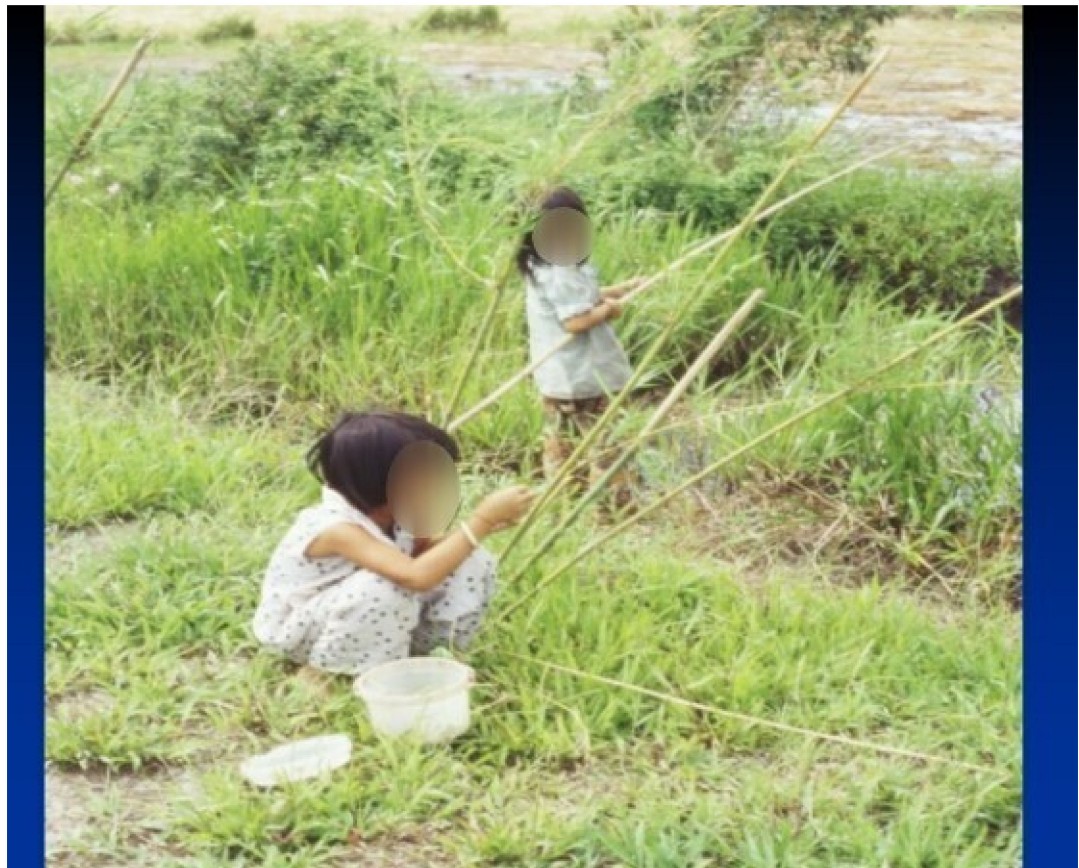

**Figure 17.** Catching even small fish in the wetlands of the Mekong Delta (Vietnam) can be an important addition to family nutrition. Source: Edward Maltby.

The emphasis on migratory waterfowl, which was the original driver of the Ramsar Convention, was seen by many observers as an indulgence on the part of richer nations that poorer countries could ill-afford [2]. Confronted with the immediate and pressing challenges of poverty, food and clean water shortages, lack of economic development and the burden of foreign debt, many developing nations were reluctant to become signatories to the Convention. They could see little advantage in undertaking management obligations perceived as primarily benefitting the conservation ethos of bird watchers in already wealthy countries. This position was to change dramatically after the 1987 Conference of Parties held in Regina, Canada [39]. Two developments were largely responsible for the change in perception and an encouragement of greater interest and commitment from the developing world.

1. Changes to the criteria for the listing of wetlands of international significance to increase the recognition of their wider functional importance beyond bird habitat;
2. Elaboration of the "wise use" commitment to emphasise the contribution of wetlands to human welfare and sustainable development.

A summary table of the changes in criteria from the Conference of Parties in Cagliari in 1980 and Montreux in 1989, either side of the Regina Conference, is given in [2]. Not all delegates at Regina initially supported the changes, with some concern that they might change responsibility for the Convention from government departments of nature conservation to those dealing with natural resources (E. Maltby personal observation).

Revision of the criteria was influenced by a rapidly growing body of scientific evidence revealing the significance of the hydrological and biogeochemical as well as ecological functioning of wetlands. These translated to benefits such as flood control, water quality and fisheries support to which people could more easily relate. Modifications to the criteria focused on how wetland ecosystems functioned, rather than their habitat description and ecological characterization orientated towards traditional nature conservation values such as rarity, uniqueness, and particularly good examples of an ecosystem type. The switch in emphasis from habitat and ecological research to ecosystem processes and functioning was led from the United States where the significance of "wetland functions and values" was linked to state wetland regulatory laws starting in 1963 [40] and from 1972 at the Federal scale to the requirements of section 404 of the Clean Water Act [41].

Internationally, the World Wide Fund for Nature (WWF) and World Conservation Union (IUCN) initiated the Wetland Conservation Programme 1985–1987. The programme set up a scientific advisory group, which *inter alia* provided technical advice and guidance to the Ramsar Convention. This included developing the first definition of "wise use" [42] and active participation in influential discussions at the Regina COP in which the present author served as chair of the session of contracting parties reviewing criteria for designating sites of international importance. The Programme also commissioned the preparation of a new text to draw international attention to the real importance of wetlands to Society. *Waterlogged Wealth, Why waste the world's wet places?* [15] attempted to capture the essence of the changing wetland paradigm, which had moved to a new emphasis beyond fundamental ecology to functioning which underpinned a wide range of benefits enjoyed by human communities.

One indicator of the influence of this change in emphasis is found in the number of developing countries signing up to the Ramsar Convention. Pre-Regina there were only 13 developing countries out of 37 signatories. Post-Regina the numbers had changed to 123 out of 167 in 2018 and are currently 124 out of 170. The proportional change from 35% to 73% is a clear reflection of the fundamental change in understanding and perception by governments worldwide that wetland conservation and sound management was not just for the benefit of the populations of rich nations, but was an important tool to underpin the improved well-being of developing nation communities.

*3.1. Elaboration of the Wise Use Concept*

Signatories to the Ramsar Convention are required to nominate at least one wetland of international importance but also undertake to promote the "wise use" of all wetlands

within their territories. The original 1987 definition was formulated by the IUCN Wetlands Programme Advisory Committee: "The wise use of wetlands is their sustainable utilization for the benefit of mankind in a way compatible with the maintenance of the natural properties of the ecosystem". Sustainable utilization was defined as "human use of a wetland so that it may yield the greatest continuous benefit to present generations while maintaining its potential to meet the needs and aspirations of future generations" [42]. Clarification and fuller interpretation of the wise use obligation coincided with the revision of the criteria for recognition of wetlands of international importance. The Regina COP can be regarded as a key milestone in the global wetland research and policy shifts in emphasis including from birds to people, habitat to ecosystems as well as nature conservation to environmental quality and human well-being.

A wide gap has often existed between accepting the definition and obligations of wise use and applying its principles and objectives to specific wetlands, or in particular countries. Maltby [43] indicated that in general the application of wise use would require:

1.  Identification of wetland functions and values;
2.  Integration of compatible uses where possible;
3.  Separation of incompatible uses;
4.  Zoning and environmental planning;
5.  Catchment management;
6.  Appropriate employment, social and economic strategies to relieve the ecosystem of damaging pressures.

The continued loss and degradation of wetlands across all signatory countries to Ramsar brings into question the effectiveness, not only of the wise use obligation but of the Convention itself, in securing the world's wetland resources. Whilst regrettably ineffective in preventing wetland loss globally, the Convention has been a key factor in raising the importance and political awareness of iconic wetlands, which are symbolic of the wider wetland resource. Such has been the case of the Florida Everglades threatened by water quality issues and hydrological changes resulting largely from agricultural development and urbanization in its wider catchment. The legal basis for intervention by the Federal Government (advised by the present author) to try and halt their progressive degradation was supported in no small measure by the triple international designation of Everglades National Park which, in addition to being a Ramsar site of international importance, is a World Heritage Site and an International Biosphere Reserve (S. Ponzoli, pers. comm.). The attorney and expert scientist fees associated with the Everglades lawsuits alone ran to many millions of dollars and the costs of remedial works orders of magnitude more. Kadlec [13] and Richardson [14] give authoritative summaries of the main issues and subsequent actions. It is unlikely that any other country in the world could afford to tackle a prosecution and mount a restoration plan on the scale seen in the Everglades. The effectiveness of such a message remains to be seen, but the over-riding conclusion is that it makes economic as well as ecological sense to avoid wetland degradation in the first place and such a strategy can be aided by following wise use guidance.

The wise use definition under the Convention was revised at the 2005 Conference of Parties (COP 9) to "Wise use of wetlands is the maintenance of their ecological character achieved through the implementation of ecosystem approaches, within the context of sustainable development". This modification reflected the influence of not only the Convention's mission statement, but also Millenium Assessment terminology [44], elaboration of the "ecosystem approach" under the Convention on Biological Diversity [45,46], and the Brundtland Commission's definition of sustainable development [47].

### 3.2. The Rise of a Functional Approach and Assessment Procedures

Concomitant with the change in emphasis from what wetlands *a* to what wetlands *do* has been the requirement to better understand and assess wetland functioning.

"A functional approach to wetland assessment is one that acknowledges that wetlands can perform work at a variety of scales in the landscape, which result in significant direct

and/or indirect benefits to people, wildlife and the environment" [48]. It effectively broadens consideration of wetlands from a view as conservation icons to recognizing their wider utilitarian importance resulting from multiple ecological, biogeochemical, and physical processes and their natural dynamics (Figure 18).

The need for a different perspective on wetlands which recognized the functions they performed (and the values resulting) was recognized initially in the United States where state wetland regulatory laws, starting in 1963, stimulated the first assessment methodologies [40]. Functional assessment subsequently became central to the US Federal permitting process, which regulates wetlands under Section 404 of the Clean Water Act (33 US Code 1344) and coupled to a "no net loss" policy. Conceptual development of assessment in the US is described by Brinson [41], and Smith [42] provides examples in practice. In developing countries, the underpinning science-based evidence is much more limited (see examples in Roggeri) [49]. Hawson et al. [50] examine approaches in Canada.

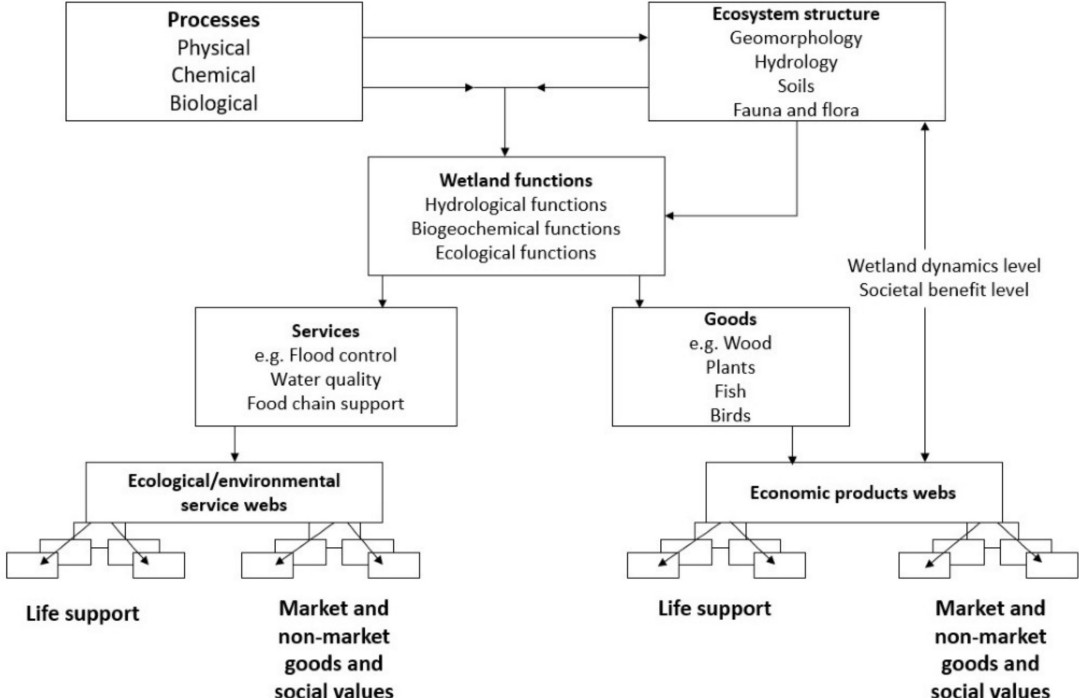

**Figure 18.** Physical, chemical, and biological processes lie behind the provision of ecosystem services [51].

The early literature from the United States was highly influential in stimulating further research, e.g., Sather and Smith [52], but whilst pioneering, was also simplistic by grouping together functions and values (without clear separation between the two) and not making sufficient distinction between the ability of different wetlands or parts of the same wetland to perform particular functions. It became increasingly clear that not all wetlands perform the same functions or perform the same functions to the same extent.

Physical, chemical and biological processes, individually or in combination, within the diverse structures of different wetland ecosystems, control different patterns and quantities of functioning such as hydrological (e.g., flood control), biogeochemical (e.g., nutrient retention) and ecological (e.g., habitat provision) functions. Functions result in the provision of different goods and services valuable to people such as flood risk reduction, pollution reduction, and food chain support (Figure 18). These and other outcomes are now commonly referred to as "ecosystem services", recognising where and how they impart human benefits.

"Assessment of functioning is a key pre-requisite to making management decisions that affect delivery, by wetlands, of specific or particular combinations of ecosystem services.

This knowledge is critical to provision of the evidence necessary to underpin strategic and policy decisions. Effective functional assessment provides an essential tool for those individuals, regulatory bodies, and other government or non-government organisations that make informed decisions for appropriate wetland management" [53].

Bartoldus [54] identified 40 different wetland assessment procedures for the United States alone, reflecting inter alia the diversity of ecosystems, species targeted, time/effort required, costs, outputs, expertise and user needs. The rationale has been driven by policy and the need to better inform decision-makers of the public values provided by wetland functioning which may be lost or impaired by development. The inevitable legal disputes involving the rights of individuals and property ownership have led to the concept in the US of "jurisdictional wetlands" with a need to delineate wetland areas (and their functions) on the basis of both legally as well as scientifically verifiable criteria [54].

Tools in the United States for "rapid" assessment have tended to treat a wetland as a single functional unit. The most widely used method initially was the Wetland Evaluation Technique (WET) developed from the work of Adamus [55] and Adamus et al. [56]. It has been the basis for training and use by the US Army Corps of Engineers (who together with the US Environmental Protection Agency are responsible for wetland permitting under the Clean Water Act legislation) and other regulatory programmes [41].

Subsequently the HGM (Hydrogeomorphic) Approach was developed to overcome criticisms of WET [57]. It was initially designed to estimate change in wetland condition by means of a quantitative comparison of altered/impacted wetlands with those that had not been altered and considered as "reference" wetlands using ecosystem functions as the basis for evaluation. Brinson [41] was intimately involved in the development of the HGM concept and provides a critical analysis of the HGM approach. He cautions on two major limitations in practice. First, it "does not provide decision-makers with complete information to determine the full consequences of degrading a wetland" and second, "only a rapid level of assessment is provided and the building of a reference system is expensive". For an up-to-date summary of recommended approaches in the United States see [39].

With financial support from the European Commission, an empirically based methodology of wetland functional assessment was developed based on trans-European multidisciplinary research [58]. The approach in Europe differed from that adopted in North America, partly because there was no regulatory framework for wetlands and partly because of the generally smaller scale of wetlands and the often intimate association with agricultural land use systems. The basic unit of assessment developed by the Maltby-coordinated international team was the "hydrogeomorphic unit" (HGMU), defined as "areas of homogeneous geomorphology, hydrology and/or hydrogeology, and under normal conditions, homogeneous soil/sediment" [58]. The mapping of HGMUs allows prediction of the variation in wetland process across often complex European wetland ecosystems and associated landscapes, based on the recognition of easily identifiable or measured controlling variables [58].

Its application has never been mandated by regulatory agencies in Europe because of the absence of specific wetland policy requirements. It remains as a tool available to predict the likelihood of a wetland, or part of a wetland, performing a particular function or combination of functions and resulting ecosystem services, together with assessment of the effects of wetland alteration without the need for expensive and time-consuming empirical research [58].

One important further application has been to link with Earth observational data to facilitate the mapping of wetland functions using remote sensing [59]. This is just one example of the very recent expansion of geospatial technologies such as GIS, remote sensing and spatial modelling that are increasingly used for wetland applications including mapping, assessment and simulations.

*3.3. Re-Discovery of the Natural Capital Value of Wetlands through Economic Determinations*

Society today ultimately is no less dependent on the globe's wetland resources than were the prehistoric communities whose livelihoods were inextricably linked with these ecosystems. Such realization has emerged at least in part from the idea of "natural capital", which recognizes the importance of the natural assets of the planet and their underpinning of economic development as well as wider human well-being. Costanza et al. [60] emphasized the disproportionate contribution of wetlands to the natural capital provided by the world's ecosystems. One estimate is that only 1% of global terrestrial rainfall flows through wetlands but the yields of ecosystem goods and services are disproportionately higher that this [60–63]. Hence, wetlands multiply the value of rainfall compared with other natural and man-made ecosystems [2]. Balmford et al. [61] concluded that conversion of wild habitats to other uses, such as mangroves to aquaculture, was always harmful in overall economic terms. They cite as one example the analysis by van Vuuren and Roy [64], who reported that for freshwater marshes in Canada, the total economic value was more than twice as high when the wetlands remained intact rather than converting them to agriculture.

The Millennium Ecosystem Assessment engaged between 2001 and 2005 over 1360 experts worldwide to assess the consequences of ecosystem change on human well-being. Their findings provide a state-of-the-art scientific appraisal of the condition and trends in the world's ecosystems and the services they provide, as well as the scientific basis for action to conserve and use them sustainably [44]. The key messages from the wetlands and water synthesis report provide insight into the condition, rate of change in the global wetland resources, and captured the new emphasis on the services provided, the value of which often far exceeded alternative land uses. (Box 1).

**Box 1.** Key messages from the MEA wetlands and water synthesis report [44].

- Wetland ecosystems (including lakes, rivers, marshes, and coastal regions to a depth of 6 m at low tide) are estimated to cover more than 1280 million hectares, an area 33% larger than the United States and 50% larger than Brazil. However, this estimate is known to under-represent many wetland types, and further data are required for some geographic regions. More than 50% of specific types of wetlands in parts of North America, Europe, Australia, and New Zealand were destroyed during the twentieth century, and many others in many parts of the world degraded.
- Wetlands deliver a wide range of ecosystem services that contribute to human well-being, such as fish and fiber, water supply, water purification, climate regulation, flood regulation, coastal protection, recreational opportunities, and, increasingly, tourism.
- When both the marketed and nonmarketed economic benefits of wetlands are included, the total economic value of unconverted wetlands is often greater than that of converted wetlands.
- A priority when making decisions that directly or indirectly influence wetlands is to ensure that information about the full range of benefits and values provided by different wetland ecosystem services is considered.
- The degradation and loss of wetlands is more rapid than that of other ecosystems. Similarly, the status of both freshwater and coastal wetland species is deteriorating faster than those of other ecosystems.
- The primary indirect drivers of degradation and loss of inland and coastal wetlands have been population growth and increasing economic development. The primary direct drivers of degradation and loss include infrastructure development, land conversion, water withdrawal, eutrophication and pollution, overharvesting and overexploitation, and the introduction of invasive alien species.
- Global climate change is expected to exacerbate the loss and degradation of many wetlands and the loss or decline of their species and to increase the incidence of vector-borne and waterborne diseases in many regions. Excessive nutrient loading is expected to become a growing threat to rivers, lakes, marshes, coastal zones, and coral reefs. Growing pressures from multiple direct drivers increase the likelihood of potentially abrupt changes in wetland ecosystems, which can be large in magnitude and difficult, expensive, or impossible to reverse.

**Box 1.** *Cont.*

- ■ The projected continued loss and degradation of wetlands will reduce the capacity of wetlands to mitigate impacts and result in further reduction in human well-being (including an increase in the prevalence of disease), especially for poorer people in lower-income countries, where technological solutions are not as readily available. At the same time, demand for many of these services (such as denitrification and flood and storm protection) will increase.
- ■ Physical and economic water scarcity and limited or reduced access to water are major challenges facing society and are key factors limiting economic development in many countries. However, many water resource developments undertaken to increase access to water have not given adequate consideration to harmful trade-offs with other services provided by wetlands.
- ■ Cross-sectoral and ecosystem-based approaches to wetland management—such as river (or lake or aquifer) basin-scale management, and integrated coastal zone management—that consider the trade-offs between different wetland ecosystem services are more likely to ensure sustainable development than many existing sectoral approaches and are critical in designing actions in support of the Millennium Development Goals.
- ■ Many of the responses designed with a primary focus on wetlands and water resources will not be sustainable or sufficient unless other indirect and direct drivers of change are addressed. These include actions to eliminate production subsidies, sustainably intensify agriculture, slow climate change, slow nutrient loading, correct market failures, encourage stakeholder participation, and increase transparency and accountability of government and private-sector decision-making.
- ■ Major policy decisions in the next decades will have to address trade-offs among current uses of wetland resources and between current and future uses. Particularly important trade-offs involve those between agricultural production and water quality, land use and biodiversity, water use and aquatic biodiversity, and current water use for irrigation and future agricultural production.
- ■ The adverse effects of climate change, such as sea level rise, coral bleaching, and changes in hydrology and in the temperature of water bodies, will lead to a reduction in the services provided by wetlands. Removing the existing pressures on wetlands and improving their resiliency is the most effective method of coping with the adverse effects of climate change. Conserving, maintaining, or rehabilitating wetland ecosystems can be a viable element to an overall climate change mitigation strategy.
- ■ The MA conceptual framework for ecosystems and human well-being provides a framework that supports the promotion and delivery of the Ramsar Convention's "wise use" concept. This enables the existing guidance provided by the Convention for the wise use of all wetlands to be expressed within the context of human wellbeing and poverty alleviation.

The cross-fertilisation between the expert guidance to the Ramsar Convention and the scientific expertise involved in the MEA was significant in ensuring that the latest thinking could be engaged within the working practices and implementation strategy of the Convention.

Hard on the heels of the MEA was the UK National Ecosystem Assessment (UK NEA)—the first ecosystem assessment to be carried out at the national scale [65]. Figure 19 depicts the Conceptual Framework of the UK NEA, and Figure 20 shows how ecosystem services are incorporated.

The UK NEA originated partly as a need to provide an evidence base on which to adopt an ecosystem approach in government policy as required by commitments to international agreements—notably the Convention on Biological Diversity as well as the Ramsar Convention. It built on the conceptual framework established by the global MEA; most importantly, it provided a direct feed into new UK government policies relating to environment and including wetlands. It aimed to provide a comprehensive overview of the state of the natural environment and a new way of expressing its wealth. The assessment was carried out under eight Broad Habitats in the UK, which meant that wetlands were considered in three of these: freshwaters including open waters, wetlands and floodplains; mountains, moorlands and heaths; and coastal margins.

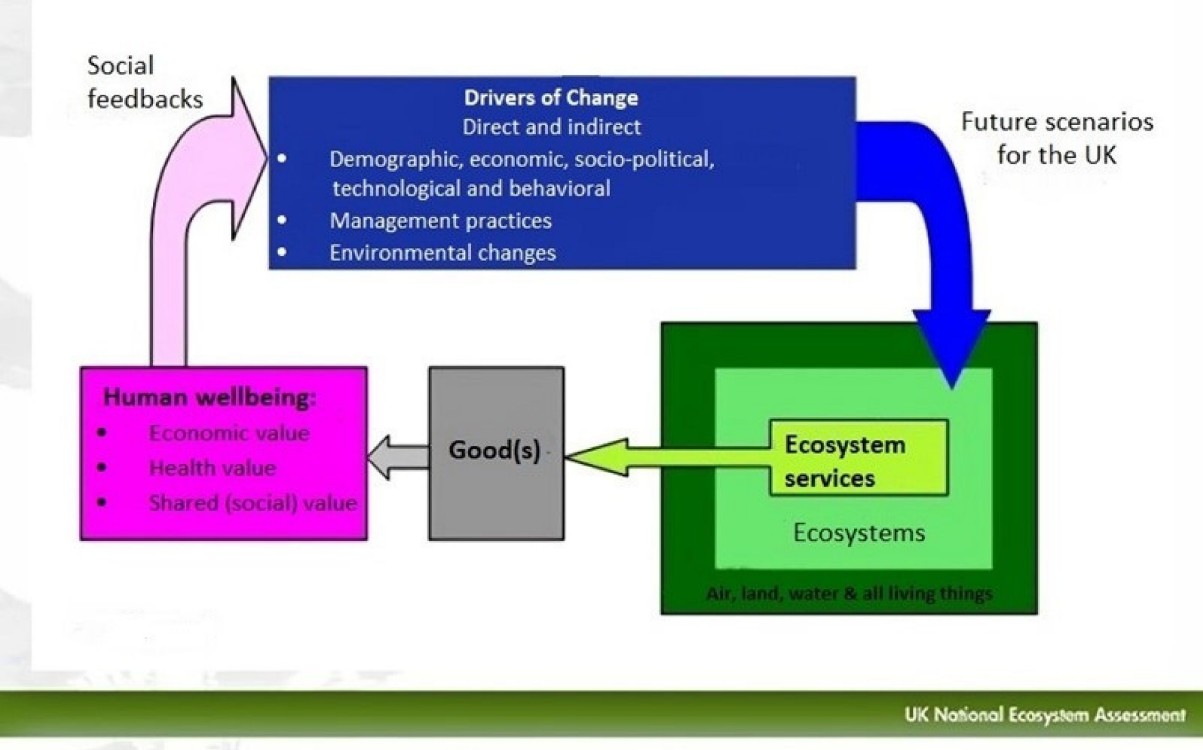

**Figure 19.** Conceptual framework of the UK National Ecosystem Assessment. Source: UK National Ecosystem Assessment 2011 UNEP-WCMC Cambridge.

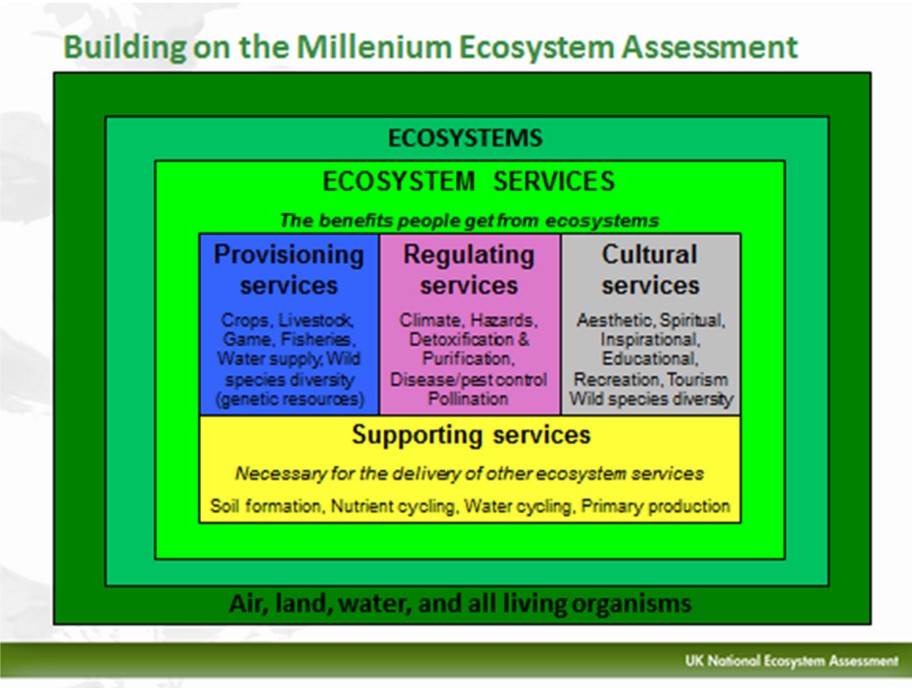

**Figure 20.** Incorporation of Ecosystem services terminology within the UK NEA. Source: UK National Ecosystem Assessment 2011 UNEP-WCMC.

Of particular significance in the assessment is that each key finding was assigned a level of scientific certainty which provides decision-makers with a level of confidence often previously lacking in the use of research findings for policy innovations. The over-arching conclusion from the freshwaters (including open waters, wetlands) and floodplains assessment was "a need to reappraise our view of the importance of Freshwater ecosystems and their critical position in policy, management and a sustainable economy. This involves recognizing the multiple benefits, potential cost-benefits and wide range of public and private interests which can be supported simultaneously . . . through a more holistic approach linked to their pivotal role in delivering ecosystem services. In turn, this recognition will arise from a more practical implementation of the ecosystem approach to integrate the sustainable management of land, water and living resources" [66], and so congruent with both the Ramsar Convention and the Convention on Biological Diversity.

The need for more holistic and integrated thinking has strongly influenced policy, at least in the UK.

The Government's 25-year Plan for the Environment sets out its comprehensive and long-term approach to protect and enhance natural landscapes and habitats. This includes working with nature and using natural capital to benefit communities.

A parallel initiative under the auspices of the Natural Capital Initiative has developed the concept of "Wholescapes", which emphasized the importance of partnership working as essential to integrate sound management of the land, its freshwater, the coast and open seas [67]. The UK National Ecosystem Assessment and follow-on Natural Capital Committee reports have catalogued the severe loss of natural capital in the UK. This is due partly to failure of land and water management practices, because individual sectors are often too narrowly focused and miss the benefits that can be achieved from working across traditional institutional and geographical boundaries. There is increasing awareness that better partnership working can support local economies, improve livelihoods, and enhance quality of life that are all consistent with meeting the objectives within the Sustainable Development Goals (particularly Goal 17 but also 11, 14 and 15). Figure 21 illustrates in a simplistic way how the "wholescapes" concept is inclusive and connective from upland freshwater catchments to the open sea and superimposes the delivery of ecosystem services on the wide range of habitats more traditionally defined.

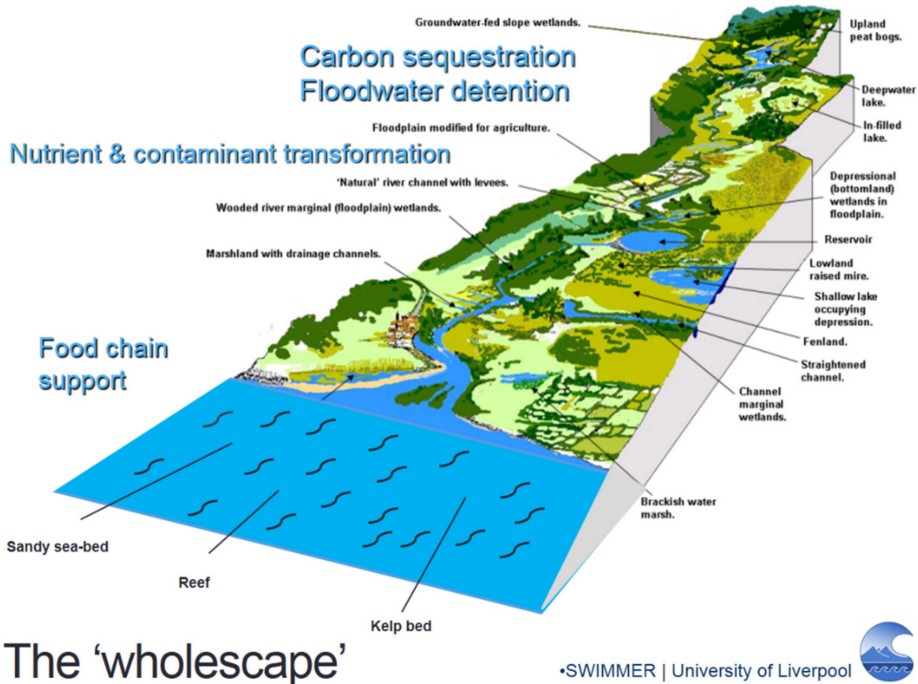

**Figure 21.** Schematic representation of the Wholescape concept [53].

The most recent changes in the focus on wetlands have concentrated particularly on their roles at scales well beyond individual sites such as within whole freshwater catchments, extensive coastal zones, major biomes and the balance of radiatively active gases in the atmosphere with a particular emphasis on their contribution to the challenges of climate change. The need to uncouple the converted uses of freshwater catchments, such as for food production or urbanization from negative impacts downstream, has brought the wide range of ecosystem services provided by wetlands into strong consideration. A new case for the conservation of peatlands based on their functioning and provision of services contributing to human well-being was outlined by Maltby [25]. Subsequently, Maltby and Acreman [67] have explained the transformation in how modern society values the benefits of natural ecosystems and highlighted the pathfinder role that wetland research has played in this paradigm shift.

Currently wetlands are at the heart of Nature-Based Solutions (NBS) to tackle issues ranging from flooding hazard and poor water quality to the many environmental challenges arising from global climate change.

There is now an unprecedented level of societal awareness of and concern over the possible far-reaching consequences of climate change. Potential impacts include increased extreme events of storms, flooding, drought and wildfires, as well as progressive sea-level rise and biodiversity loss. Wetlands can provide mitigation in respect of all these impacts as well as playing a vital role in the dynamics of greenhouse gases. An assessment of Nature-Based Solutions for Climate Change in the UK coordinated by the British Ecological Society sets out measures that can be used in "Freshwaters, Peatlands and Coastal and Marine systems" as well as other natural systems [67].

A key conclusion from the Freshwaters assessment is that NBS are best and most effectively delivered by local-based partnerships. The role of "Trusted Intermediaries" is vital to facilitate local support, attract resources, and foster engagement of local communities and facilitate Citizen Science. This finding is consistent with the success of the Rivers Trust movement throughout the UK in implementing a wide range of wetland restoration actions with the support of volunteers from the local communities.

### 3.4. Upstream and Downstream Thinking

Changing farming and land management practices aims to reduce the movement of sediments, pesticides and animal waste into rivers. In southwest England, moorland restoration is helping to achieve these aims and improve the condition of rivers, such as the Dart and Exe. This reduces downstream water treatment costs, defers large capital investments, and lowers household water bills. This "Upstream Thinking" approach has been facilitated through a partnership of South West Water, Exmoor National Park, Devon's and Cornwall's Wildlife Trusts, and the Westcountry Rivers Trust (https://wrt.org.uk/project/upstream-thinking/ (accessed on 26 March 2022)).

The long-term goal is to see partnerships amongst and between government, civil society and businesses that operate at the whole scale—linking, where appropriate, land, the coast and sea. Although wholescape is based on geography (the bio-physical scale), its application needs a transformation in human behaviour to affect a cultural change also at this scale. Successful implementation will also require the parallel application of the ecosystem approach and the new thinking that this necessitates (Figure 22).

### 3.5. The Economics of Ecosystems and Biodiversity (TEEB)

Paralleling the new emphasis on natural capital and the ecosystem services which result from wetland functioning is the recognition of the real economic values of wetland ecosystems and water. The Economics of Ecosystems and Biodiversity (TEEB) is a global initiative focused on "making nature's values visible"; its principal objective is to mainstream the values of biodiversity and ecosystem services into decision-making at all levels. It aims to achieve this goal by following a structured approach to valuation that helps decision-makers recognize the wide range of benefits provided by ecosystems and biodiversity,

demonstrate their values in economic terms and, where appropriate, capture those values in decision-making. The findings from the part of the programme of work dealing with wetlands capture the essence of the challenges facing policymakers worldwide (see Box 2).

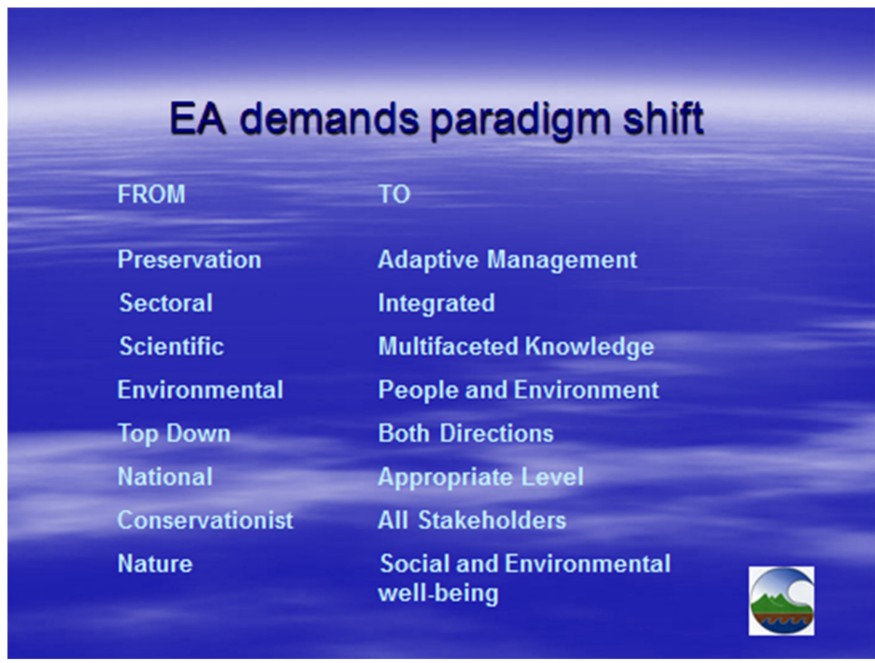

**Figure 22.** Summary of the key changes in thinking required to effectively apply the ecosystem approach [46].

**Box 2.** Key Messages from TEEB Wetlands and Water 2013.

1. The "nexus" between water, food and energy is one of the most fundamental relationships—and increasing challenges—for society.
2. Water security is a major and increasing concern in many parts of the world, including both the availability (including extreme events) and quality of water.
3. Global and local water cycle are strongly dependent on wetlands.
4. Without wetlands, the water cycle, carbon cycle and nutrient cycle would be significantly altered, mostly detrimentally. Yet policies and decisions do not sufficiently consider these interconnections and interdependencies.
5. Wetlands are solutions to water security—they provide multiple ecosystem services supporting water security as well as offering many other benefits and values to society and the economy.
6. Values of both coastal and inland wetland ecosystem services are typically higher than for other ecosystem types.
7. Wetlands provide natural infrastructure that can help meet a range of policy objectives. Beyond water availability and quality, they are invaluable in supporting climate change mitigation and adaption, support health as well as livelihoods, local development and poverty eradication.
8. Maintaining and restoring wetlands in many cases also lead to cost savings when compared to manmade infrastructure solutions.
9. Despite their values and despite the potential policy synergies, wetlands have been, and continue to be, lost, or degraded. This leads to biodiversity loss—as wetlands are some of the most biodiverse areas in the world, providing essential habitats for many species—and a loss of ecosystem services.
10. Wetland loss can lead to significant losses of human wellbeing, and have negative economic impacts on communities, countries and business, for example through exacerbating water security problems.
11. Wetlands and water-related ecosystem services need to become an integral part of water management in order to make the transition to a resource efficient, sustainable economy.
12. Action at all levels and by all stakeholders is needed if the opportunities and benefits of working with water and wetlands are to be fully realised and the consequences of continuing wetland loss appreciated and acted upon.

### 3.6. International Recognition of the Significance of Trans-Boundary Water Control on Wetlands, Political Conflict and the Effects on Human Culture and Livelihoods

One of the outcomes of the war in Iraq was media attention and increased public awareness of the vulnerability of one of the world's most ancient wetlands, and the unique Madan culture associated with the floodplain marshes and shallow water bodies of the former Mesopotamia.

The Mesopotamian marshlands are fed by the Tigris and Euphrates rivers mainly through the flood pulse generated by the annual snow melt from the mountains of Turkey, and to some extent Iran. Numerous tribes of marsh dwellers, Marsh Arabs or "Madan" provide the cultural link with Sumer, one of the world's earliest civilisations. In a pioneering interdisciplinary international study, funded by the NGO Amar Appeal, attention was drawn to the importance of the wetland ecosystems in supporting the livelihoods as well as cultural traditions of the Madan [68,69]. Their significance also at the regional and wider international scale was illustrated in terms of the connectivity to important bird flyways, linkages of aquatic species between the marshes and the Gulf, and as home for rare and endemic species [68,70].

Until recently the marshes occupied an area of some 25,000 km$^2$ but about 90% of the wetlands existing in the 1970s had disappeared by 2000 [69]. Further assessment in 2003 indicated that just 7% of the original area remained (Figure 23). It was estimated that the entire wetland would disappear within 10–20 years [68], and in 2003 by UNEP within 5 years.

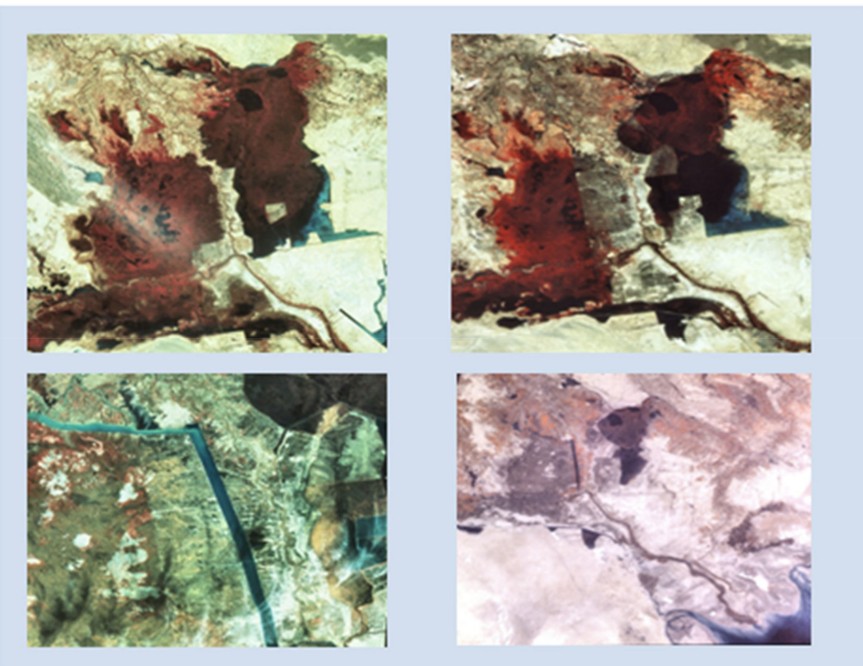

**Figure 23.** Landsat imagery of the Mesopotamian marshes showing the progressive desiccation from the 1970s to 2000 [68,69].

Dam construction and other engineering works upstream in Turkey and Iran, as well as Iraq itself, provided the Saddam regime with the opportunity to accelerate the drainage of the marshlands through deliberate diversions within the wetlands themselves. Driven by the desire to suppress the traditional opposition from Madan tribes and the provision of refugia for political opponents, the Saddam administration may have displaced up to half a million people, with many killed and the remainder becoming "environmental refugees" [69]. The impact, both on human misery as well as wildlife and cultural heritage, was seized upon by the global media and politicians creating strong public awareness and engagement of national and international NGOs as well as governments.

The uncoordinated restoration of the marshes started in 2003, very soon after the fall of the Saddam regime. Individual communities took the initiative to breach local structures that were preventing normal flooding and wetland regeneration. More organized restoration of hydrological flows followed, and in Spring 2005 UNEP reported recovery of more than 50% of the marshland present in the 1970s (see https://postconflict.unep.ch/publications/UNEP_IMOS.pdf (accessed on 30 March 2022)). There was a remarkably swift re-establishment of many species [71,72], reinforcing the adage "just add water and nature will take care of the rest" (Figures 24 and 25).

The post-war Iraq government, new NGOs such as Nature Iraq and the international aid community (especially US AID) invested heavily in planning and assessment initiatives. From personal observations in the field and more recent reports of others, the restoration of what has been regarded as a "human and environmental catastrophe" is technically feasible [2,73]. Most recently Crisp [73] describes occurrence of the deepest water conditions in the marshes for some 20 years due in part to the increased release of Tigris and Euphrates flows from Turkey together with higher rainfall. In response, there has been significant recovery of important bird and fish species. More than 250,000 people have returned since 2003 but most have occupied surrounding towns and villages rather than the reed houses and floating islands associated with the traditional Madan culture.

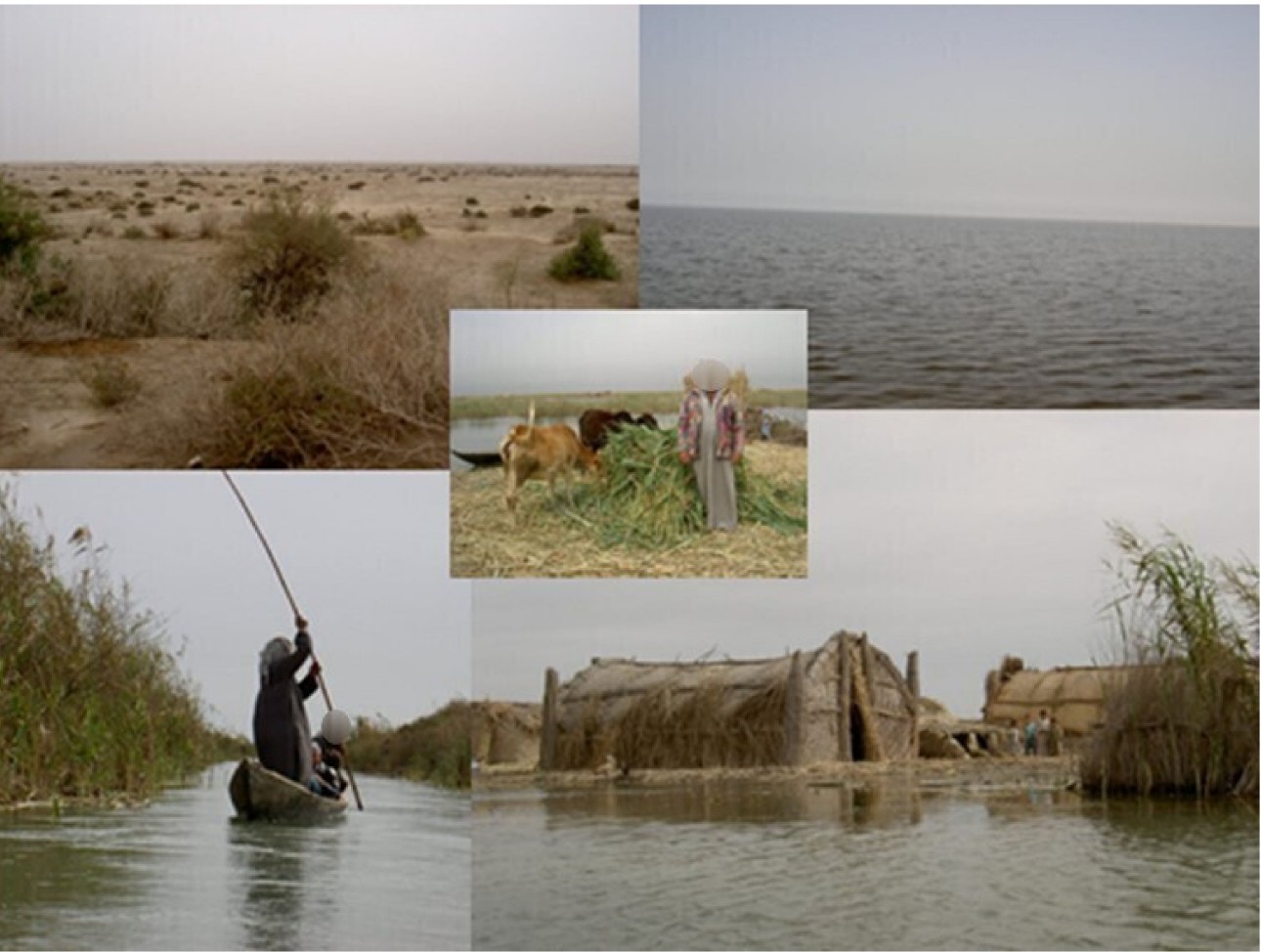

**Figure 24.** Mesapotamian wetlands. Desiccated landscape. Reflooded impoundment with limited biodiversity. Restored wetland within the natural floodplain landscape with traditional canoe, reed housing and animal husbandry. Source: Edward Maltby.

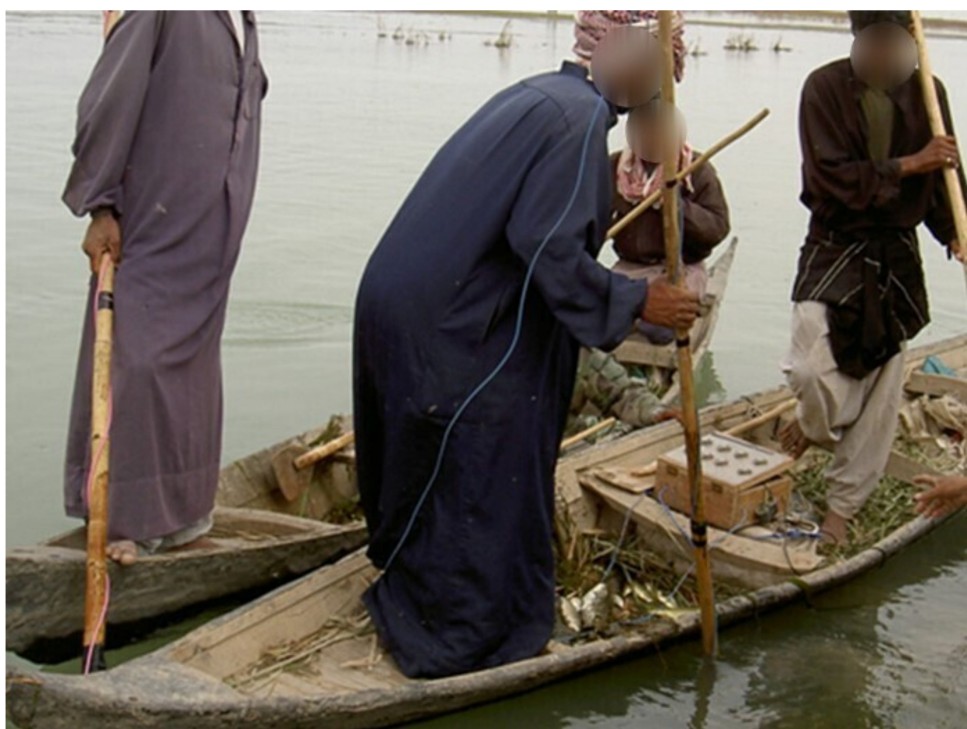

**Figure 25.** Electrofishing in restored Mesopotamian marshland often replaces more traditional methods, and without sound management and community cooperation may threaten the sustainability of stocks. Source: Edward Maltby.

This likelihood was forecast in interviews carried out by this author during the US AID funded international mission in 2004. There is no doubting the importance of wetland restoration in the opinion of many of the local inhabitants to recover the historic land and waterscape as well as support some livelihoods, environmental quality, and biodiversity. However, there is wide consensus that this should not deny people access to improved health care, more formal education, better transport facilities and energy resources.

The challenge is to achieve the most appropriate balance of restored "natural" ecosystems and sustainable development with the support of the local population, together with awareness of its significance to the wider regional and interests. This may require innovative financial mechanisms to compensate or incentivise communities, in a way like the shift in policy in the UK and elsewhere towards payment for ecosystem services, recognizing that their provision to the benefit of others generally comes at a cost to the local custodians or owners of the resource.

Sustainability of the right balance will inevitably require progressive river basin dialogue among Turkey, Iran and Iraq. It will require comprehensive engagement of diverse stakeholders with often conflicting demands for water and other resources in the region (Figure 26).

The principles of the Ecosystem Approach offer guidance on how to achieve an integrated solution to maintaining sustainable land, water and living resources [47]. There is a potentially important role for the Ramsar Convention to catalyse the ongoing process.

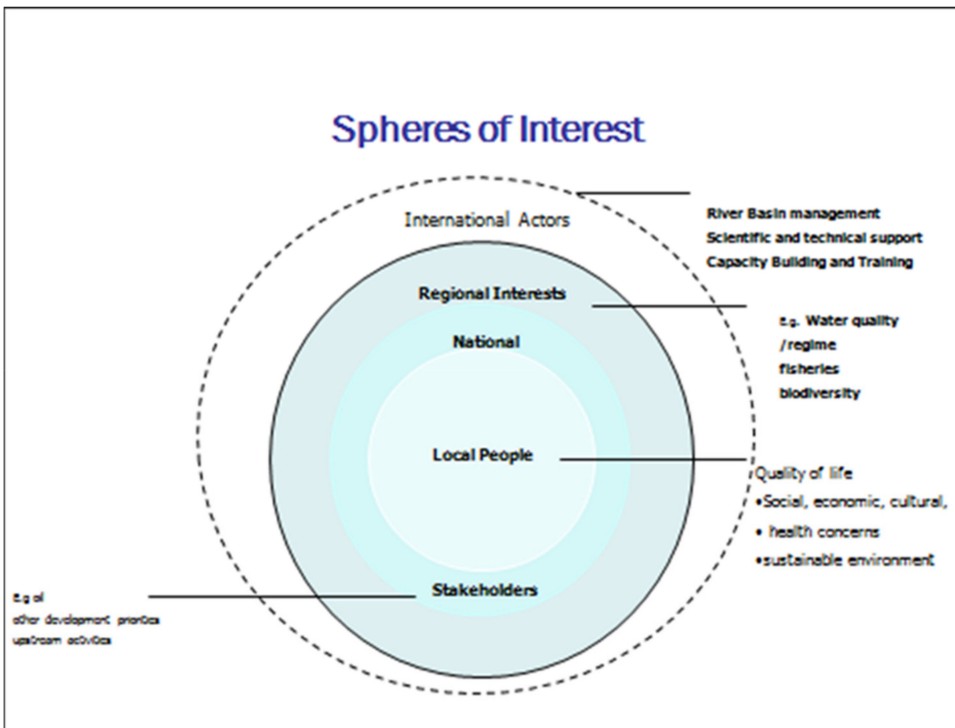

**Figure 26.** Spheres of influence of stakeholders as presented by the author to the round table discussions among Iraq, Iran and international delegates at the UN Geneva 2003 [74].

*3.7. Increase in the Political Priority of Wetland Restoration*

A significant element of the paradigm shift in wetland management in recent years has been an increase in momentum of wetland restoration at a range of scales and for a variety of purposes. The overriding driver for the restoration effort has been recognition that such restoration is required to underpin economic and social well-being.

Notably in the United States, The Comprehensive Everglades Restoration Plan (CERP) was authorized by Congress in 2000 as a plan to "restore, preserve, and protect the south Florida ecosystem while providing for other water-related needs of the region, including water supply and flood protection". At a cost of more than USD 10.5 billion and with a 35+ year timeline, this is the largest hydrologic restoration project ever undertaken in the United States, if not the world [75]. It is a further irony that the need for such restoration is to support the population growth, urban expansion, and economic development that have actually been responsible for reduction of the historic extent of Everglades. The Water Resources Development Act (2007) further authorized funds towards restoration of the Florida Everglades. It also recommended the establishment of a Coastal Louisiana Ecosystem Protection and Restoration Task Force that would make recommendations and propose strategies to protect, repair, restore and maintain the ecosystems of the Louisiana coastal zone. This built on The Coastal Wetlands Planning, Protection and Restoration Act, (CWPPRA), the federal legislation enacted in 1990 that is designed to identify, prepare, and fund the construction of coastal wetlands restoration projects. Since its inception, 210 coastal restoration or protection projects have been authorized, benefiting approximately 100,000 acres in Louisiana [76]. Federal, State and university partners have dedicated considerable research effort into the understanding of the underlying causes of coastal wetland loss in Louisiana, currently estimated at about 75 square kilometers per year, and determining the optimum solutions to mitigate and/or reverse such dramatic losses and the resulting economic and human misery [77]. There has been considerable scientific debate over the most sustainable approach to restoration and management. This has focussed in large measure on the restoration of sediment nourishment to the Mississippi Delta. The review by Allison and Meselhe provides some insight into the range of scientific

opinion [78]. Current approaches at individual project locations include one or more of the following actions: marsh creation and restoration, shoreline protection, hydrologic restoration, beneficial use of dredged material, terracing, sediment trapping, vegetative planting, barrier island restoration, and bank stabilization techniques.

It is particularly noteworthy that in recent years China has made considerable advances in wetland conservation and protection which now includes over half of the nation's extensive resource. The *People's Daily* newspaper reports wetland expansion of more than 200,000 hectares from 2016–2020, and a new wetland protection system comprising mainly national and more local wetland parks and wetland nature reserves [79]. The enhanced status of wetlands is enshrined in the Wetland Protection Law, passed by the national People's Congress in December 2021 and which entered into force in June 2022 [80]. The new law includes a specific section on wetland restoration in which the principle "natural restoration first" is emphasised partly in recognition that some restoration projects have actually caused more ecological harm. In addressing how to make the law work in practice, Wang Xinyi and Sheng Xiaoying from Friends of Nature, have stressed the importance of public participation in making the law work, especially because of the crucial knowledge of local communities. The potential benefits in creating jobs and prosperity to local residents is a powerful argument for sound management, and where necessary wetland restoration. Almost 10,000 villagers close to the Longji terraced fields national wetland park in South China's Guangxi Zhuang autonomous region have benefitted from Park revenue and local government subsidies. According to the National Forestry and Grassland Administration (NFGA) China's wetland parks contributed CNY 53.6 billion (USD 8.21 billion) to regional economic growth and created 47,000 new jobs. In 2019 China's national wetland parks received 385 million visitors [79].

In Europe, the most recent five year environmental report [81] portrays a somewhat dismal picture of the state of the continent's waters, with only 40% of surface water bodies achieving good ecological status and wetlands widely degraded; this is especially the case for floodplains, resulting in a "critical impact on the conservation status of wetland habitats and the species that depend on them". The report strongly advocates a river basin scale and ecosystem-based approach to future management, with restoration including natural water retention measures and buffer strips to parallel economic measures such as smart water pricing, more efficient irrigation, and so-called precision agriculture. Restoration of wetlands is a common theme supported by policy at the European and individual country levels with the objective of reconnecting rivers to their floodplains a high priority [82,83].

## 4. Structure of a New Paradigm

An attempt is made in Figure 27 to capture the complexity of the new perspectives of wetlands amidst the present challenges facing Society. The paradigm shift, which is now merging nature, economic growth and prosperity within the same conceptual framework, is being embraced increasingly in policy initiatives worldwide.

### 4.1. Current Priorities and Future Challenges

Civil society faces arguably unparalleled new threats that may lead to severe and accelerated degradation of our remaining wetland resources but could also offer great opportunities for wetlands to play a renewed role in sound planetary management. The nature of these threats can be grouped under the headings of **F**ood, **E**nergy, **W**ater, **E**nvironment (FEWE). Associated with all of these is the global biodiversity crisis which is a fundamental constraint to actually achieving sustainable economies, livelihoods and human well-being. To what extent can wetlands be seen as the nexus of the threats and opportunities associated with **FEWE**?

Human conflicts, plus forced and unforced migration, are creating refugee crises at various scales and have raised public as well as political awareness of the stark reality and far-reaching implications of climate change; these are just some expressions of exacerbated concerns for sustained human well-being.

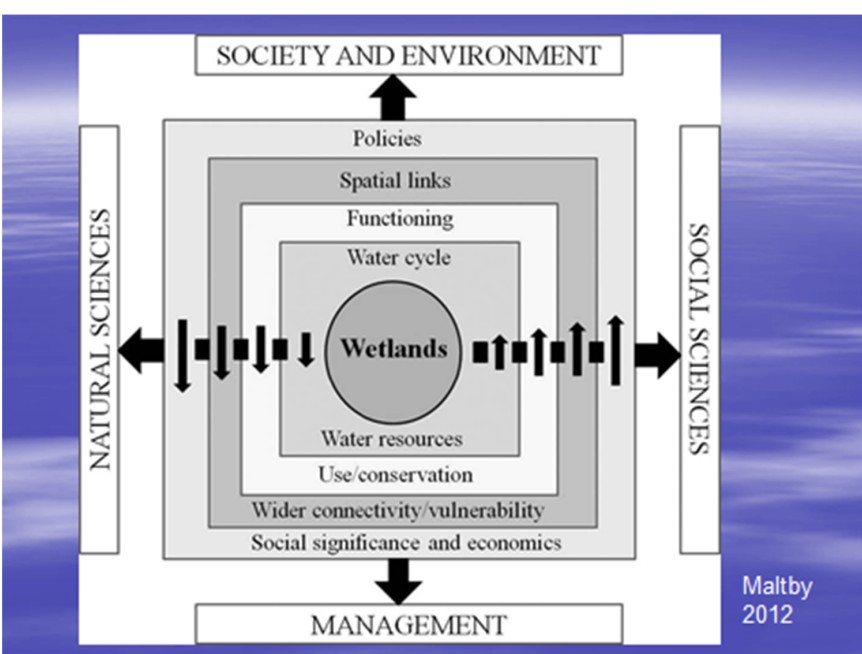

**Figure 27.** Outline of the new wetland paradigm with multiple linkages among the natural world and human society [2].

Current food shortages and dramatic price inflation arising from the war in Ukraine are likely to renew pressures to convert wetlands and other natural ecosystems to agriculture. A notable historical precedent and example of such a threat comes from the expansion of soybean cultivation in the southern United States from the late 1950s. Between 1959 and 1964, 400,000 hectares in the Mississippi Delta region were drained and cleared almost exclusively for soybean, and in north Louisiana alone forested wetlands disappeared at 45,000 hectares a year [15]. The economics of wetland forest clearance and drainage were given an unanticipated twist in the early 1970s by events off the Pacific coast of South America. Years of over-fishing were compounded in 1972 by a change in the upwelling pattern of the cold, nutrient-laden Peru current resulting in collapse of the anchovy industry, one of the most important sources of protein meal. The demand for soybean increased as a replacement and prices soared. Despite the enormous costs of clearance and drainage, the anchovy crash made cultivation of even difficult wetland terrain financially attractive. Between 1973 and 1975 over 200,000 hectares of wetlands on the coastal plain of North Carolina were cleared. Large agricultural corporations with substantial outside investment converted enormous tracts of intact pocosin(literally 'swamp on a hill') wetland into agriculture. The pocosins once covered almost a million hectares of North Carolina, but by 1980 only 21,800 hectares remained with considerable, but then largely ignored, loss of wetland ecosystem services [84]. Maltby [15] records numerous other examples where changing economics and policy decisions such as relating to taxation (as was the case with commercial afforestation of the peat bogs of the Scottish Flow country) [37], have led to dramatic wetland losses.

At the present time there are major concerns for the global supply of sunflower oil because of the conflict in Ukraine. It remains to be seen if this results in renewed pressures for the conversion of more peat swamp forests in Southeast Asia to palm oil plantations, where already deep concerns surround existing rates of expansion and the loss of habitat for iconic species such as the Orang Utan together with wider ecosystem services [85] (Figure 28).

There is little doubt that the world food and other manufacturers will be closely monitoring the need for alternative ingredients or new country sources, should supplies from Ukraine and elsewhere continue to dwindle and prices rise further. Developing

countries, in particular, facing mounting international debt and possibly famine, may find it difficult to resist the pressures which may lead to further wetland degradation and loss in the bid to realize short-term economic returns and meet the problem of food commodity shortages.

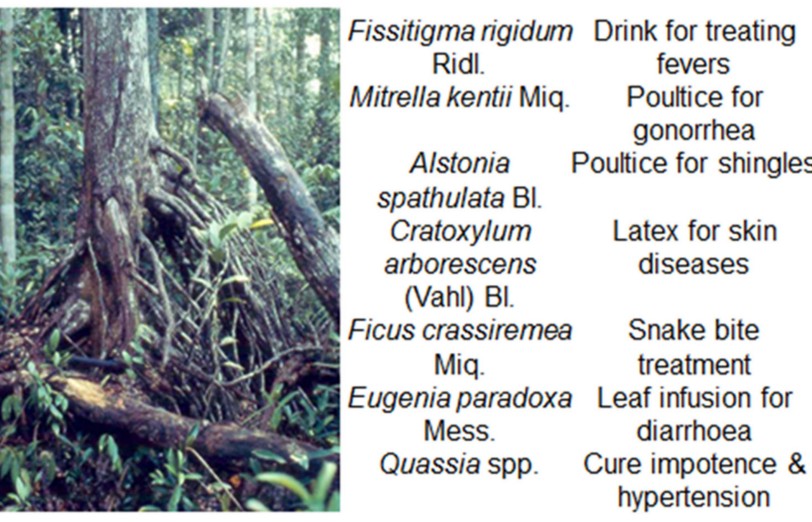

**Figure 28.** An illustration of the frequently ignored services provided by Peat Swamp Forests in Southeast Asia. After Chai et al. [86]. Photograph Edward Maltby.

Even if wetlands are not converted directly to support additional food production, intensification of farming will inevitably result in greater pressures on catchment hydrology, especially downstream water quality and biodiversity resulting in adverse changes in ecological character and wider ecosystem functioning. The dramatic extent of potential off-site impacts has been well demonstrated in the case of the Everglades and has been a major concern of the Ramsar Convention (Figures 29 and 30).

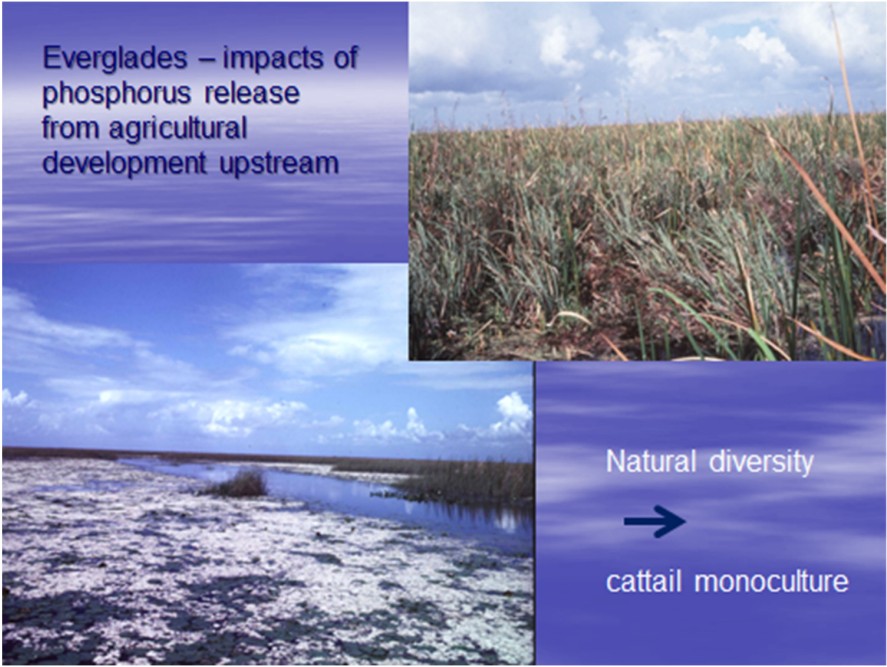

**Figure 29.** Wetland stresses and impacts on ecosystem structure and functioning in Everglades. Source: Edward Maltby.

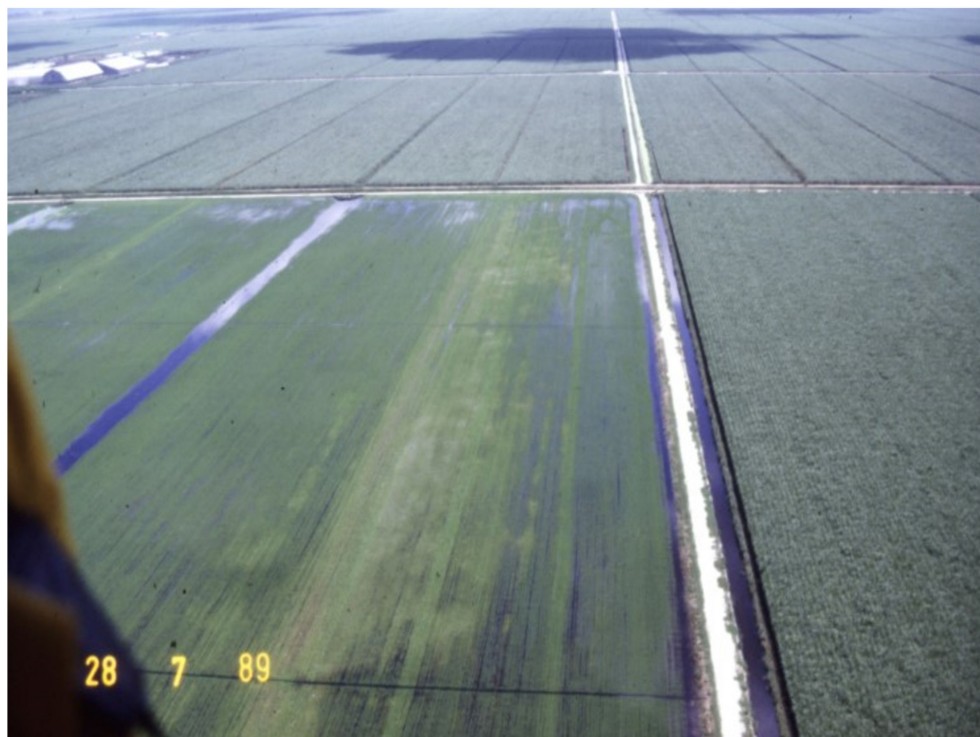

**Figure 30.** Wetland conversion to agriculture outside Everglades National Park—the Everglades Agricultural Area. Source: Edward Maltby.

It is possible that the use of modified wetlands, such as Storm Treatment Areas (STAs), aimed to reduce the nutrient load from the Everglades Agricultural Area (Figure 30) reaching and altering the wetland ecosystems of the Everglades National Park, can be a potential solution but at costs that may prove prohibitive for application elsewhere. At smaller scales, wetlands may be used effectively as buffer zones to protect adjacent aquatic ecosystems from pollution and ecological damage [87]. Whilst policy has often encouraged the use of buffer zones immediately adjacent the water bodies to be protected, such locations may not be the most effective because of the pattern of runoff and nutrient pollution pathways (Figure 31).

Whilst wetland buffer zones may be highly effective in reducing nutrient pollution and protecting water quality, there may be a downside contribution to global warming resulting from the incomplete process of denitrification releasing nitrous oxide rather than nitrogen gas (Figure 32). Decisions to use wetlands control of nutrient pollution will involve an assessment of "trade-offs" and cost-benefit analysis that goes well beyond short-term economic expedients. This will always depend on the availability of an empirically sound and verifiable evidence base to which the scientific community has contributed already a great deal, but will benefit from increasingly targeted policy-relevant research.

In the face of such major threats to wetlands it is even more important to emphasise the many ways in which these ecosystems actually contribute to the wider sustainable supply of food—a key part of the provisioning services they provide.

It may be necessary now to follow the example set by the IPCC reports, and assemble the evidence regarding the benefits from the healthy functioning of the wetland resource and the consequences of their loss and/or degradation, in such a way that policy makers within governments can take a much more informed, urgent and serious view of what is at stake. The Ramsar Convention could reasonably take a strong lead in such an initiative and subsequent "Wetland Days" may assist in generating supportive public awareness, building on already highly relevant themes of previous years.

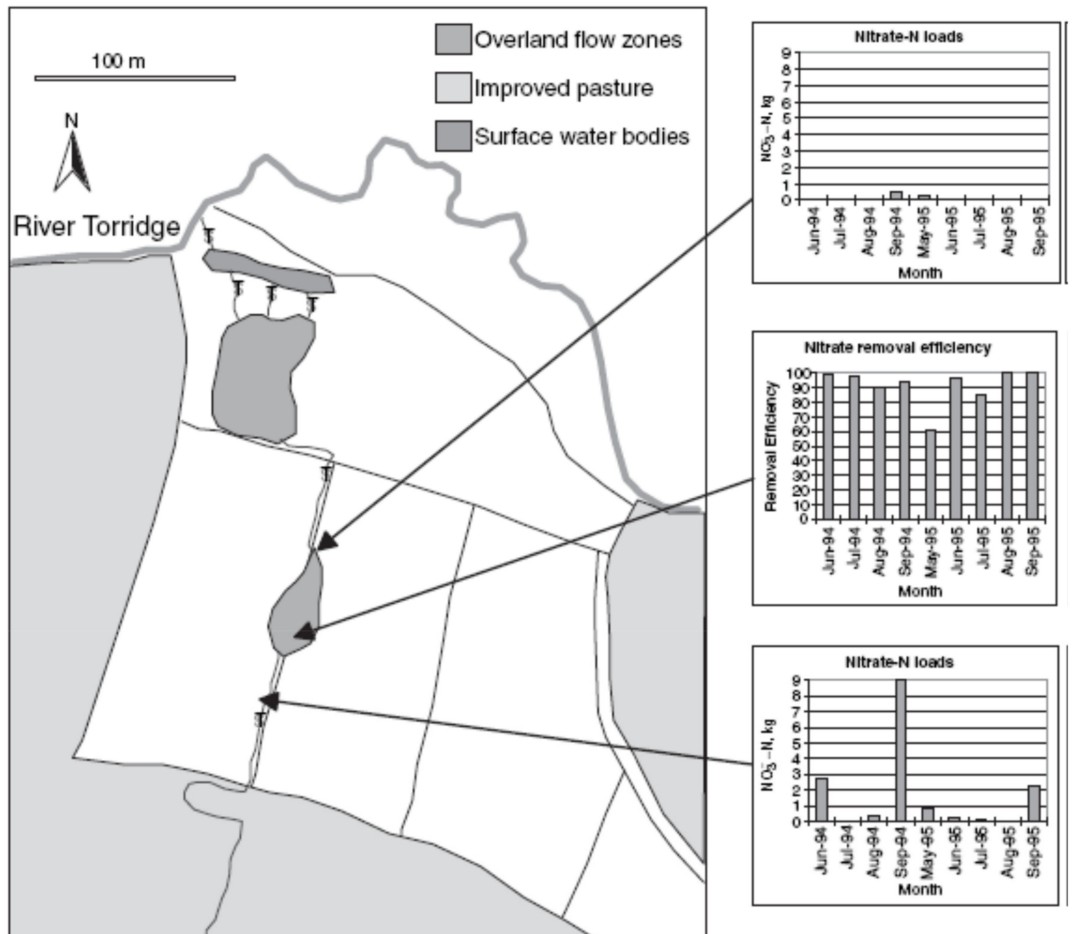

**Figure 31.** Incongruity between wetland science and policy in which river marginal wetlands running parallel to the channel may not necessarily offer the most effective buffers against nitrate pollution of freshwaters. The graphs show reduced levels of nitrate at locations along the runoff flow line from the source in agricultural fields and at right angles to the alignment of the channel. The grey shaded areas are particularly effective wetland hotspots for nitrate reduction especially by denitrification [88].

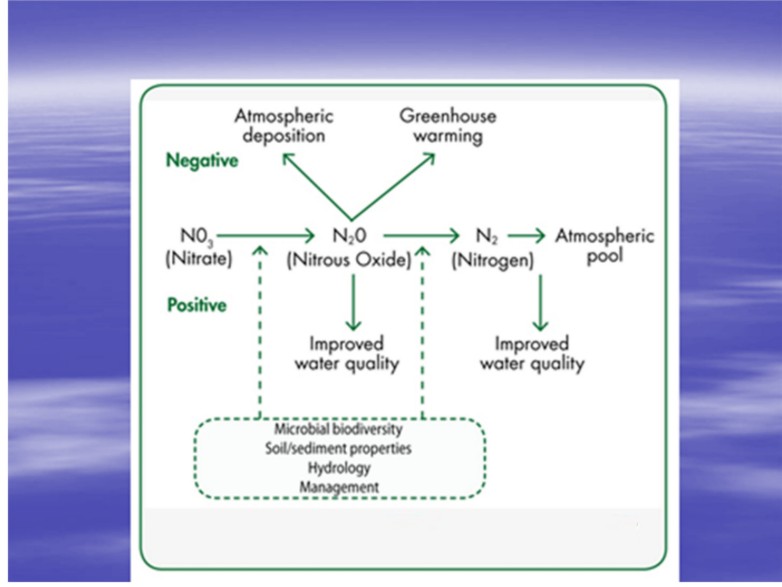

**Figure 32.** Example of trade-offs between water quality maintenance and global warming [89].

*4.2. Energy*

The rationale for a drive away from hydrocarbons and the development of alternative renewable energy sources has become generally well-accepted in the face of increasingly incontrovertible evidence for their role in global warming. The argument has gained additional force because of the 2022 spike in world energy prices and an increasing imperative for nations to become less dependent on potentially less-reliable supplies from other countries.

The switch to "green" energy sources is of paramount importance, but care must be exercised where there is the possibility of wetland loss and/or degradation that could outweigh any gains. This might be the case in inappropriate tidal barrage schemes, hydropower dams, or selecting wetland areas for solar or wind power if potentially adverse impacts on ecosystem functioning are either ignored or subjugated to a lesser importance.

*4.3. Water*

The world has a finite freshwater resource, and the lack of adequate, safe and reliable supply is a major contributor to poverty and deprivation for a significant proportion of the global population. Wetlands are a vital link in the hydrological cycle, controlling flows as well as water quality and mediating the essential connectivity from atmosphere through the land to the sea. Modification of hydrology has been an ever-increasing feature of human history. There are increasing examples of the restoration of pre-existing hydrological conditions including reconnecting rivers to their floodplains [82], managed realignment to accommodate rising sea-levels, and re-wetting peatlands. It is unlikely, however, that future pressures which result in over-exploitation of water resources will cease. In addition to increasing populations, urbanization, and industrial and agricultural pressures, the premium in real estate values attracted by water-side locations will continue to threaten river, lake and coastal wetlands even though in many countries such new development is now severely restricted [90].

Over-exploitation of groundwaters has been a long-standing threat to iconic wetlands dependent on those aquifers, and the Tablas de Damiel in Spain offers a salutary example. Despite its designated and "protected" status, the wetland was severely degraded by groundwater abstraction from the surrounding cultivated area via the La Mancha aquifer in the 1970s. Desiccation combined with high temperatures led to combustion of the peat substrate and extensive fire damage to the ecosystem. The lesson is that even wetland "jewels" included within a nation's conservation network are not necessarily safeguarded from external damaging demands for water. Llamas 1988 [91] describes the conflicts over water use and still the emergency remedial measures to save the wetland are on-going in the face of limited financial resources. Additionally, over-use of coastal aquifers by increasing urban populations, as well as sea-level rise, are causes of salt-water intrusion which can extend far inland resulting in ecological as well as economic damage [92,93].

Desiccation of major wetlands such as Lake Chad in Africa [94] and the Aral Sea in Central Asia [49], in addition to the case of Mesopotamia, are clear signposts to the reality of potentially more extensive losses due to poor management of water resources.

Regional rainfall and hydrology will be affected by climate change. Increases in extremes and unpredictability of droughts and floods are likely consequences, and in this case, wetlands can be an invaluable resource adding a high level of natural resilience to freshwater catchments against the adverse consequences of climate change. Particular benefits will include cost effective management of flood peaks and maintenance of base flows.

*4.4. Environmental Change and Health*

Concerns surrounding climate change have brought into sharp focus the importance of the natural environment and its "natural capital" in mitigating, arresting or even reversing the effects and/or rate of global warming. Wetlands offer some of the most potent tools to meet the challenges posed by the current climate crisis. Of particular importance is their

role as sinks or stores of carbon, which otherwise would add to the atmospheric load; these are well-documented.

Somewhat less explored are the physical and mental health benefits to be gained from engagement with the wetland ecosystem. There is at least some evidence that investment in improving access to wetlands may be cost effective in relation to the returns in human health benefits and reduced costs to traditional health care providers [95].

Responding to the so-called climate crisis in terms of the wetland resource is not invariably straightforward, and an area of concern must be the potential for an increased threat from disease vectors either in type or geographical extent. The complexity and difficulties in predicting the effects of climate change on important disease vectors have been examined by Rocklov and Dubrow [96]. Analysis is urgently required of the potential role of wetlands in facilitating the increased risk of water-borne or other disease vectors under various climate change scenarios and strategies developed for prevention or amelioration.

### 4.5. World Wetlands Day and the Future

The themes of a quarter of a century of World Wetlands Day, celebrated in this special series, well reflect the new wetland paradigm in which human well-being is a central focus. This in no way reduces the importance of biodiversity and the migratory birds which were the original primary focus of the Ramsar Convention. Biodiversity is a good indicator of ecosystem health, which in turn reflects the ability to function in ways that can deliver the services essential for sustainable livelihoods. Healthy wetland ecosystems are an important part of Earth's natural capital.

## 5. A New World Charter—Conclusions

Despite the increasing evidence base for the vital natural and economically important roles played by wetlands, the resource continues to decline and degrade. The pressures on wetlands imposed by Society's responses to current and possible future social, economic, and environmental challenges are unlikely to diminish. Frustrated by the lack of effective policies to protect wetlands, a transdisciplinary team have proposed a Universal Declaration of the rights of Wetlands [97]. (Box 3).

**Box 3.** The proposed Universal Declaration of the rights of Wetlands.

| | |
|---|---|
| 1. | The right to exist |
| 2. | The right to their ecologically determined location in the landscape |
| 3. | The right to natural, connected, and sustainable hydrological regimes |
| 4. | The right to ecologically sustainable climatic conditions |
| 5. | The right to have naturally occurring biodiversity, free of introduced or invasive species that disrupt their ecological integrity |
| 6. | The right to integrity of structure, function, evolutionary processes and the ability to fulfil natural ecological roles in the Earth's processes |
| 7. | The right to be free from pollution and degradation |
| 8. | The right to regeneration and restoration. |

They conclude by urging "all governments, from local to national, as well as international organisations to support this Declaration and provide mechanisms and funding for implementation and enactment". They "specifically encourage the Contracting Parties (countries) to the Ramsar Convention on Wetlands to seek ways to embrace the Declaration and to incorporate the rights of wetlands into their national procedures and operational processes".

Such an initiative is to be applauded, but we must remember that it is people who use and abuse wetlands and it will be at the individual or community level that their values or "rights" will be safeguarded for future generations and the welfare of the planet. Wetland scientists can contribute significantly to this effort, and the ways in which this might be achieved are summarized in Figure 33.

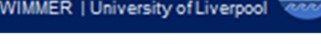

**Figure 33.** Some Key Challenges for wetland science. Source: Edward Maltby.

Human evolution, prehistoric community survival and the development of civilisations owe much to the natural capital provided by wetlands. Failure to maintain recognition of such significance led to the progressive demise of these ecosystems worldwide, and throughout history. Whilst the rise of conservation ethics and the emergence of ecology as a respected scientific discipline did much to underpin their importance, it was the Ramsar Convention that was of key significance in raising awareness internationally. Notwithstanding the role of Ramsar as a standard bearer of the wetland conservation movement, it has been a renewed recognition of the importance of their natural capital, manifest as wide-ranging ecosystem services, that has reshaped the perspective of Society. Realisation of the roles of wetlands in underpinning human well-being has come full-circle and is the essence of the paradigm shift for wetland scientists, managers and policy-makers alike.

**Funding:** This research received no external funding.

**Informed Consent Statement:** Informed consent was obtained from all subjects whose identifiable image appears in the figures.

**Data Availability Statement:** There are no data sets to support this review.

**Acknowledgments:** The author would like to thank the many wetland science colleagues worldwide who have generously shared knowledge, insight and friendship that has made his own scientific journey so rewarding. The research referred to would have been impossible without the support of my family and successive teams of research assistants and students together with a plethora of funding bodies who showed confidence in their investment. The present contribution has benefitted from helpful recommendations from anonymous reviewers and the editorial staff of the Journal have provided exceptional and much appreciated support in finalizing the manuscript.

**Conflicts of Interest:** The author declares no conflict of interest.

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
