# Peer review of "The Wetlands Paradigm Shift in Response to Changing Societal Priorities: A Reflective Review"

_land, doi:10.3390/land11091526_

Round 1

Reviewer 1 Report

The manuscript presented for review concerns the issue of the wetlands paradigm shift in response to changing societal priorities. This is a review manuscript. Overall, the manuscript is written in an interesting and relatively comprehensive manner. The topic is up to date. However, I have some comments on the text. In my opinion, for a review manuscript, the text contains too few citations of the literature. There are whole passages of the text without quoting the literature (e.g. First Stson Introduction) where it begs to add citations. Additionally, Fig. 8; 9; 10; 18; 19 are not legible and require improvement. The text needs to be edited, editing errors appear, eg} instead of) etc. Some figs look like presentation slides - they require better quality.

Author Response

I appreciate the helpful comments.

The whole field of wetland science is too large for the review to cover all aspects. I have modified the title to reflect the fact that this is very much related to the author's direct experience of the evolution of the science and policy framework over the last half century. The apparent gaps in citations generally reflect areas that are either commonly accepted knowledge or are well covered elsewhere. I have attempted to improve the figures and in the case of the ppt slides I am hoping that the editorial process can modify these to an acceptable level. I have gone through the entire text to correct obvious errors.

Reviewer 2 Report

This paper is a well-written document, providing insightful reviews of the history of wetland conservation and management in response to changing societal priorities.  The paper addresses a variety of wetland issues in the context of ecosystem management, the Ramsar convention, and regulatory assessment methods among many others. The materials used in the paper are rich and relevant. Its readability is high.

While the manuscript should be accepted for publishing in Land, it could be further enhanced if the author would like to include brief reviews of the following topics, focusing on new developments of wetland science and management:

(1)         Wetland restoration. This is an important part of the paradigm shift. This review paper should give at least one short paragraph to discuss it. Particular examples may include South Florida (Everglades) Ecosystem Restoration under the Water Resources Development Act and Coastal Louisiana Wetlands Restoration under the Coastal Wetlands Planning, Protection and Restoration Act. These have been the largest wetland restoration projects in the United States and probably in the world. 

(2)         Major wetland conservation efforts in other parts of the world. The author may consider giving brief reviews about the European Union overarching environmental policies in the context of possible conflict with its constituent countries’ efforts on wetlands. Over the past 20 years or so, China has made significant efforts to protect or rebuild its wetland resources including establishing many national wetland parks and urban wetland parks. Recently China passed its first wetland legislation, Wetland Protection Law, effective on June 1, 2022. The author may consider making brief comments on some of major wetland-related efforts in China. 

(3)         Geospatial methodology for wetland applications. In recent decades, geospatial technologies, such as GIS, remote sensing, and spatial modeling, have been increasingly used for wetland applications like mapping, assessment, and simulation. There have been numerous publications in these areas. Even a short paragraph of summary reviews would greatly help connecting this paper to related research and management communities.

For this review paper (not original research), the above are the recommendations to the author, not required revisions.

Author Response

I appreciate the very helpful comments and especially the suggestions related to (1) wetland restoration- I have an additional section on this important topic which is also addressed incidentally in various other parts of the mss.

(2) major wetalnd conservation efforts in other parts of the world- I have introduced significant new material from the United States, Europe but also especially China where new initiatives are highly relevant.

(3) geospatial methodology for wetland application-I have covered this now with additional material related to the linkages to functional assessment.

Such suggestions have helped to improve the mss.

Reviewer 3 Report

Review of Maltby, E. (Land):

The Wetlands Paradigm Shift in response to Changing Societal  Priorities: A Review

General Comments:

Although admittedly a “self-indulgent examination” this lengthy manuscript is not a much a review of the literature but instead an essay o the author’s thoughts regarding wetlands within a social construct. Unfortunately, the author has not presented a clear outline of the essay’s thesis. To this end, it is difficult to read not knowing where it will lead.  This reader suggest the manuscript be re-constructed.

Specific Comment:

The Abstract us only 77 words long.  For abstracting service, it would be preferential to provide more details (the specific conclusions reached, why this review is needed, contributions since other similar reviews (if applicable), etc.

Line 22. Not “and/or” but “and”; in most instances “and” is preferable.

Line 24. “…seminal role in Earth history”?  Rather broad statement that shod be toned down considering the more recent historical reference in the following statement (Ramsar).

Line 38. “that” not “which”.

Line 48. “Figure 1” not “Figure 2”.

No reference to Figures 2, 3, and 4, is made in the text.

Line 56: continental drift; line 84 “aquatic ape”…; line 111 Flood of Noah

Figure 7. Difficult to view the text; pay attention o accessibility in the use of colour (https://www.ascb.org/science-news/how-to-make-scientific-figures-accessible-to-readers-with-color-blindness/)

Fi8. 8&9: layout is poor; they do not provide rich information.

Fig. 10.  There are far more than 3 types of freshwater wetlands; this figure is not informative

I have stopped making specific comments since the paper will likely require significant modifications to its text in order to be re-organized and more coherent.

Author Response

I appreciate the reviewer's critical comments.

I have modified the title to reflect the somewhat selective nature of the review which concentrates on the personal experience and perspective of involvement in wetland science and conservation effort over half a century. It is hoped that this will be of some interest to the wider and especially younger generation of scientists and managers.

The abstract has been lengthened considerably to accommodate the concerns expressed by the reviewer.

The specific comments have been addressed with the exception of modifying the assertion  that wetlands have played a seminal role in Earth history. I stand by this view and welcome any further debate that this might generate.